# Aging and memory of transitional turbulence

**Vasudevan Mukund, Chaitanya S. Paranjape, Michael Philip Sitte [●],
Gökhan Yalnız [●] & Björn Hof [●] [✉]**

The recent classification of the onset of turbulence as a directed percolation (DP) phase transition has been applied to all major shear flows including pipe, channel, Couette and boundary layer flows. A cornerstone of the DP analogy is the memoryless (Poisson) property of turbulent sites. We here show that, for the classic case of channel flow, neither the decay nor the proliferation of turbulent stripes is memoryless. As demonstrated by a standard analysis of the respective survival curves, isolated channel stripes, in the immediate vicinity of the critical point, age. Consequently, the one to one mapping between turbulent stripes and active DP-sites is not fulfilled in this low Reynolds number regime. In addition, the interpretation of turbulence as a chaotic saddle with supertransient properties, the basis of recent theoretical progress, does not apply to individual localized stripes. The discrepancy between channel flow and the transition models established for pipe and Couette flow, illustrates that seemingly minor geometrical differences between flows can give rise to instabilities and growth mechanisms that fundamentally alter the nature of the transition to turbulence.

Many stochastic processes are memoryless, i.e., the probability of an event to occur is constant in time, as is the case for radioactive decay. Fluid flows, however, are governed by deterministic equations, and in principle, the flow state at an instant in time fully determines the entire future evolution. It may therefore come as a surprise that the transition to turbulence in a large number of flows is governed by memoryless processes. This seemingly contradictory state of affairs is reconciled by the sensitive dependence on initial conditions—a defining property of deterministic chaos that equally applies to turbulent fluid motion. The associated exponential amplification of even minute differences makes long term forecasts of the dynamics impossible in practice, and effectively erases the flow's memory.

It is precisely this memoryless property that has paved the way for a possible solution to the century-old puzzle surrounding the nature of the transition to turbulence. More specifically, the transition in Couette, pipe and related shear flows is characterized by the co-existence of laminar and turbulent regions. Such patches of turbulence, although frequently long-lived, are transient, and their decay is abrupt and unpredictable—more precisely, memoryless. The underlying dynamical-systems model is that of a chaotic saddle in phase space, characterized by an exponential lifetime distribution[1,2], i.e., a constant escape rate[3], independent of age. This correspondence to a chaotic saddle[2,4–14] is probably the best established property of transitional turbulence in subcritical shear flows, and a large body of literature has been dedicated to the dynamical-systems interpretation, including studies of the underlying exact coherent structures[1,15–17], the boundary crisis[18–22] that gives rise to transient chaos, and the resulting edge state that mediates the eventual escape from turbulence[23,24]. Corresponding transient dynamics have equally been observed for turbulence in astrophysical accretion discs[25], in nonlinear fibre optics[26], for the dynamo transition in magnetohydrodynamic turbulence[27], turbulence in pulsatile[28] (e.g., cardiovascular) flows and surprisingly even extend to the generic case of isotropic turbulence[13].

In addition to the temporal dynamics, also the spatial proliferation, i.e., the seeding of new patches via a contact process (called 'puff splitting'[29] in pipes), turned out to be memoryless[30]. These combined insights have not only allowed the determination of the critical point for the onset of sustained turbulence[30], but also confirmed an earlier

Institute of Science and Technology Austria (ISTA), Klosterneuburg, Austria. ✉e-mail: bhof@ist.ac.at

more general conjecture by Pomeau[31] that the spreading of turbulent fronts may be connected to directed percolation (DP). Unlike in the original proposition, active sites were found to correspond to macroscopic patches (puff or stripes) of turbulence, which, just like in models of DP, can either decay or infect their neighbors[30], and the probability for either event to take place does not depend on time. Based on these insights, DP is believed to be the standard route to turbulence for this entire class of flows[32–39].

At first sight, channel flow perfectly complies with all of the above criteria. In the transitional regime, turbulence appears in the form of stripes, and simulations of such stripes in periodic boxes confirmed the memoryless nature of decay and splitting of individual stripes[14,40,41]. Like in other shear flows[30,32], the critical point has been approximated by comparing mean life and splitting times[40]. However, for channel flow these results have been obtained in numerical simulations of flows in relatively small boxes with periodic boundary conditions. Here stripes cannot take their natural, localized form but instead the stripe ends merge via the periodic boundaries. While it is not clear how such artificial conditions affect the transition in channel flow, studies in Couette flow found memoryless lifetime statistics and qualitatively similar proliferation dynamics for both periodic[32] and fully localized stripes[42]. Studies of localized channel stripes in large domains[43–46] were usually limited to very few realizations, and consequently a statistical analysis of neither stripe decay nor stripe proliferation has been conducted. At the same time analogies to decay and proliferation of transitional structures in pipe and Couette flow are typically taken for granted and memoryless stripe dynamics are assumed (e.g., Manneville and Shimizu[46]).

We will show in the following that stripe localization fundamentally alters the proliferation and decay processes of turbulence. Unlike for other shear flows, channel stripes become sustained via a periodic (and hence deterministic) growth process, which involves the persistent generation of streaks at the downstream tip. Below this threshold, stripes can live for long times but eventually decay. Even for characteristic time scales exceeding those of clearly memoryless cases in other flows by an order of magnitude, the lifetimes of channel stripes are not exponentially distributed. Following the standard classification of survival functions, stripe decay instead is an aging process. In this case, the probability of a stripe to decay/die increases in time. We further show that above the critical point, with size, stripes are increasingly likely to fracture into two or more pieces, a process that intermittently gives rise to growing daughter stripes. Surprisingly, also this stripe reproduction mechanism has an increasing hazard rate and corresponds to an aging process.

## Results

Experiments are carried out in a large aspect ratio channel, considerably exceeding domain sizes of most earlier studies. Specifically, the channel's dimensions are $(L_x, L_y, L_z) = (4000h, 2h, 490h)$, where $L_x$, $L_y$ and $L_z$ are lengths in the streamwise, wall-normal and spanwise directions, respectively. Quantities are non-dimensionalized with the length scale $h$ (half-gap) and 1.5 times the bulk velocity $U_b$ (where $1.5U_b$ corresponds to the centreline velocity in case of laminar flow). In keeping with numerical and theoretical studies, we define the Reynolds number as $Re = 1.5U_b h/\nu$, where $\nu$ is the kinematic viscosity. Unless perturbed, the flow stays laminar over the entire Reynolds number range investigated. We used two protocols to obtain isolated stripes. Stripes were created either by perturbing the flow across a certain width ($\approx 20h$ in the spanwise direction) or a stripe was thus created at a higher Re, followed by a quench to the target Re. The latter mechanism was found to be more efficient at lower Re, at which the former creates stripes only sporadically. At Re where both mechanisms work well, results are qualitatively the same regardless of the perturbation type, provided that the created stripes are of comparable size.

## Onset of stripe expansion

Turbulent stripes are investigated in the range $600 \le Re \le 900$. For each Re, around 1000 individual stripes are generated near the channel entrance. As they advect downstream, they are monitored by multiple cameras along the channel until they eventually exit the setup. More details are available in the Methods section. Given that stripes travel with a speed close to the bulk velocity, the total observation time permitted in our $4000h$ long channel (following convention, time is non-dimensionalized by $h/(1.5U_b)$) corresponds to 6000 advective units. Stripes are typically allowed to develop over the first $1500h$ (measured from the inlet) and measurements focus on the downstream region (see the Methods section for details). In all the experiments reported in the present study, stripes remained at a distance of at least $50h$ from the lateral boundaries to avoid interactions, stripe advection velocities and tip speeds remain unchanged up to this distance. Overall more than 10,000 stripes are tracked in order to obtain reliable statistics and the total observation time exceeds $5 \times 10^7$ advective time units. The resulting images are analysed to extract various quantities of interest. In particular, the area occupied by turbulence is determined as a function of time and used to compute the mean growth rate of turbulence (see the Methods section for details). Note that this analysis is independent of the actual proliferation mechanisms (e.g., stripe extension, broadening, stripe splitting, branching etc.) and only determines if the overall fraction occupied by turbulence either increases or decreases. For each Re -1000 measurements were carried out, starting from individual stripes of varying length (stripe length varied between $120h$ and $240h$ approximately, see SI Fig. S1 for typical stripe size distributions), hence sampling a wide range of different initial conditions. In this context it has been shown previously in simulations[44] of isolated stripes that growth rates (as well as the stripe survival) depended on the initial stripe length if lengths were $\lesssim 110h$ but became length independent for larger values and this effect hence does not play a role for the stripes considered in the present study. The growth rates obtained in our experiments are shown in Fig. 1. For Re$\lesssim 650$, growth rates are negative, while above this Re, the growth rate becomes positive and continues to increase with Re.

We propose that once individual stripes proliferate also turbulence as a whole will eventually cease to decay, and hence the critical point for channel flow can be estimated to be Re$_c \approx 650$. This estimate neglects stripe interactions and more precisely provides a lower bound for the critical point (see also Avila et al.[30], Hof[39] for analogous arguments). In any case, our experiments suggest that the critical point is considerably lower than that reported by Sano and Tamai[35], an observation that is in qualitative agreement with direct numerical simulations (DNS) of channel flow, where persistent stripe growth has been reported from Re $\approx 660$[44,47], a value slightly larger than the experimental estimate. Another recent numerical study[45] estimated a value of Re $\approx 700$ for the onset of sustained stripes, but it is noteworthy that here a different Reynolds number definition has been used, which is based on the centreline velocity of laminar flow of equal frictional drag, and therefore in the presence of turbulence, generally results in higher values.

## Stripe lifetimes and categories of survival curves

As is customary for turbulent shear flows, we investigate the transient nature of turbulent structures below the critical point via a lifetime analysis. At a given Reynolds number, we hence determine the stripes' survival probability as a function of time. Based on ample studies in shear flows, the resulting survival curves are expected to have exponential tails, as is characteristic for a memoryless, non-aging process.

However, more generally, survival curves can fall into different categories; in population ecology three standard types are commonly distinguished[48]. As illustrated in Fig. 2a, for type 1, the decay rate is

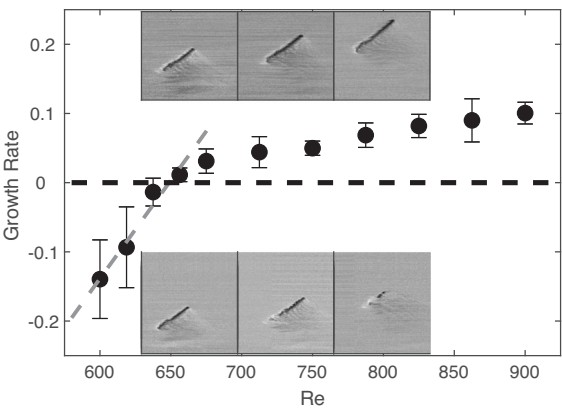

**Fig. 1 | Transition from shrinking to expanding turbulent stripes.** The mean growth rate of turbulent regions is plotted as a function of Re. Error bars indicate the standard deviation across different experimental runs. The first 4 points are fitted by a straight line to estimate the Re at which the growth rate first becomes positive, which is Re ≈ 650. Above this Re, individual stripes grow, while below, they eventually decay, as seen in the insets showing a sequence of flow visualization images.

initially zero and increases with time, which corresponds to the standard aging process (positive aging). Type 2 is the previously discussed memoryless case with a constant hazard rate. Type 3 conversely is characterized by a decreasing hazard rate and is also referred to as negative aging. All three cases are generally modeled by Weibull distributions, where the survival probability is given by

$$S(t) = e^{-(t/\tau)^k} . \tag{1}$$

Type 1 is defined by $k > 1$, type 2 by $k = 1$ (the exponential distribution) and type 3 by $k < 1$. The characteristic time scale $\tau$ typically increases rapidly (superexponentially) with increasing Reynolds number. The expected qualitative trend with increasing Re is indicated in Fig. 2a for the customary type 2 memoryless case and for the contrasting type 1 aging case.

The measured survivor curves for channel flow are shown in Fig. 2b for three Reynolds numbers, 600, 620 and 630. Based on the estimated value of the critical point in our experiment, $Re_c ≈ 650$, this corresponds to reduced Reynolds numbers (relative distance to the critical point) $\epsilon = (Re - Re_c)/Re_c$ of −8%, −5% and −3%. For comparison, we show three survival curves measured previously for pipe flow[11] at −15%, −9% and −8% from critical. While as expected, the pipe curves (red) are of type 2 and hence non-aging (memoryless), for channel flow (shown in black) this is surprisingly not the case. Despite having longer characteristic lifetimes (see Table 1), which underline the proximity to the critical point, the survivor curves are of type 1: fitting Eq. (1) we obtain $k$ values ranging from 6 to 9 (see Table 1 for details), marking stripe decay as an aging process. When stripes are young, their survival probability is essentially one and decays are absent. As stripes grow older, the first decays are encountered until at late times, the survival probability drops faster than exponential, and the remaining survivors decay in rapid succession, which practically sets a maximum life expectancy. As the critical point is approached the characteristic lifetime increases (see Table 1).

The circumstance that young stripes do not decay makes it practically impossible (given experimental constraints) to determine characteristic lifetimes for Re > 630, i.e., even closer to the critical point. This is in stark contrast to pipe flow where excessively long lifetimes can be obtained in comparably short pipes using censored data. In such cases single puffs are not monitored from birth to death but instead typically just for a few hundred advective time units (e.g., Peixinho and Mullin[6], Kuik et al.[49]), shorter than the several thousand

advective time units realized in the present study. While data censoring works extremely well for memoryless cases, it practically fails for type 1 survivor curves.

## Stripe lifetimes and decay in direct numerical simulations

The survival curves measured in our channel experiment not only fundamentally differ from those in pipe and Couette flows[7,32], but surprisingly also from computations of channel flow in small domains. As discussed above, stripes in such small computational boxes cannot fully localize and are found to be memoryless[14,40]. An example of such non-aging channel stripes[40] is shown in Fig. 2c (red data points). The Weibull fit (Eq. (1)), like in the case of the pipe distributions (Fig. 2b), results in a value of $k$ close to 1 (see Table 1 for details).

Given the fundamental difference to the experimental distributions (Fig. 2b), and that no comparable numerical study exists for fully localized stripes, we conducted DNS at Re = 620 in a much larger domain size of $(400h, 2h, 400h)$. Here stripes fully localize and assume their natural shape. As shown in Fig. 2c (blue circles), stripe localization causes survival curves to change from type 2 to type 1. The Weibull fit (Eq. (1)) results in a value of $k$ close to 9, defining it as an aging process and confirming the experimental finding. While the simulations are qualitatively in excellent agreement with the experiments, small shifts in absolute values remain, which may be attributed to the differences in boundary conditions.

In order to obtain some insight into the origin of stripe aging, we consider the time evolution of the stripes' turbulent kinetic energy, shown in the upper panel of Fig. 2d. In this case, the energy gradually decreases with time. This is qualitatively different from memoryless cases, such as puffs or localized stripes in Couette flow[42], where the energy is expected to fluctuate around a constant mean during the lifetime of turbulence until the decay sets in and the energy sharply drops. An example of the latter memoryless case is shown in the lower panel of Fig. 2d (light grey curves) for periodic stripes computed for the same domain size as in Gomé et al.[40] at Re = 750. Here, decays can occur with equal probability at any time and the eventual decay process takes ~200 advective time units.

Following recent studies[50–52] we reanalyse the fully localized stripes and decompose the velocity field into small and large scale flows (see SI for details). The small scales correspond to the turbulent fluctuations (streaks and streamwise vortices), while the large-scale flow corresponds to a circulation around the stripe, which has more recently been shown to drive an inflectional instability at the stripe's downstream tip[53]. Whereas the large-scale contribution decreases continuously (see SI Fig. S3), the small-scale flow, when normalized by the stripe area (see curves in color in the lower panel of Fig. 2d), initially fluctuates around a constant level until finally the viscous decay sets in. This final steep decay of energy takes about 200 advective time units, and hence matches the eventual collapse of the periodic memoryless stripes (light grey curves in same panel). These observations suggest that the small scales, i.e., the local vortices and streaks that make up turbulence, remain intact during the stripe's lifetime (as should be the case). Conversely, the decreasing magnitude of the large scale (SI Fig. S3) is associated with the aging process.

## Tip instability and streak generation

Instead of energy, which is not readily measurable in experiments, we next consider the stripe size and position as illustrated in Fig. 3. A peculiar feature of channel stripes is that they do not simply advect downstream but they move diagonally across the channel. This spanwise motion is caused by the aforementioned instability at the stripe's downstream tip[53] and the resulting continuous creation of streaks, see Fig. 3c and Supplementary Movie for illustrations of this process. Unlike for growth processes in other shear flows, in particular unlike for stripes in Couette flow[54,55], here streaks are created at a constant rate (Fig. 3d) and this process is hence deterministic. From our

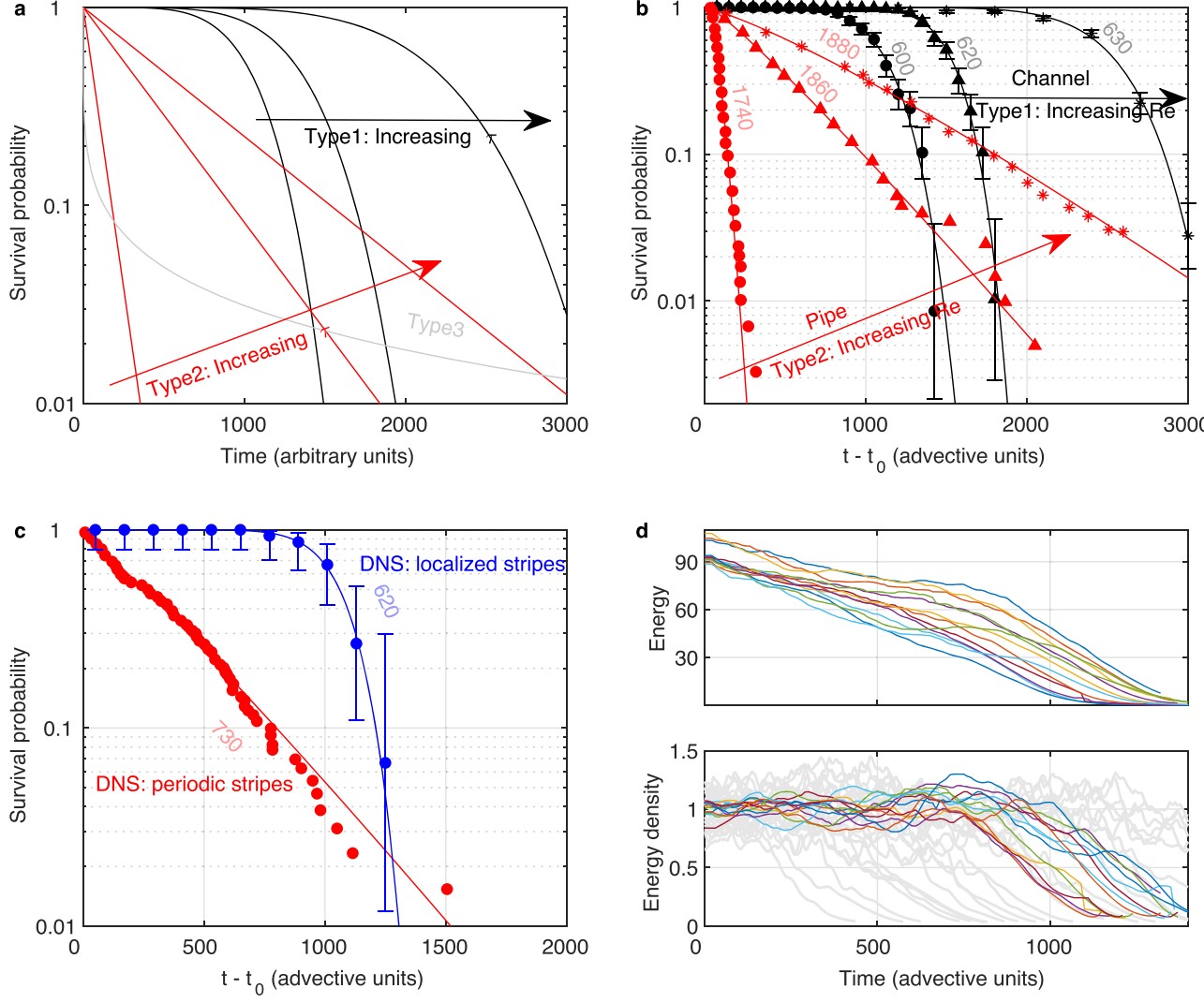

**Fig. 2 | Types of survival functions. a** The three types of survival curves, type 1 (positive) aging, $k > 1$ in eq. (1). Here the decay rate is initially zero and monotonically increases in time. Type 2 non-aging, $k = 1$. The decay rate is constant. Type 3 negative aging, $k < 1$. The decay rate is largest at $t = 0$ and decreases with time. Arrows indicate the trend for increasing $\tau$ for type 1 and type 2, the cases relevant to the present study. **b** Survival probabilities for the channel stripes in experiments are given by the black symbols for Re = 600, 620 and 630. Error bars indicate 95% CI. In all cases the data is accurately matched by a Weibull fit (Eq. (1)) and the resulting $k$ values are considerably larger than 1 (see Table 1), marking stripe decay as a type 1 aging process. For comparison three survival curves for puffs[11] are shown in red. In this case, as expected the Weibull fit results in values of $k$ that are close to one confirming the memoryless nature of puffs. **c** Comparisons between survivor curves in channel simulations. In red for periodic stripes computed in small domains, where $k$ is close

to one and hence the decay memoryless, and in blue for fully localized stripes computed in large domains where $k \approx 8$. Error bars indicate 95% CI. Hence, while periodic channel stripes are of type 2, non-aging, localized stripes are of type 1, aging. **d** Evolution of turbulent kinetic energy of stripes in DNS. The top panel shows the total kinetic energy which continuously decreases in time. The bottom panel shows the energy of the small scale flows (see SI), normalized by the stripe area. This quantity reflects the local magnitude of streaks and vortices, which fluctuate around a constant value while the stripe is alive and eventually drop when the stripe decays. For the memoryless periodic stripes in light grey (computed at Re=750) it is the total energy (shown per unit area) that fluctuates around a constant mean, and no decomposition is needed in this case. In both cases the average value of the respective plateaus has been set to one.

experiments, it becomes apparent that above the critical point (Fig. 3a at Re = 655) the streak creation and hence the stripe's spanwise speed is constant; below critical (Fig. 3b at Re = 620) this process slows down. During the initial phase, i.e., while the stripe is young, the streak creation is fully intact and streaks are created at the same rate as for the sustained case (compare blue dashed lines in Fig. 3a,b). However, at the lower Reynolds number, at some point the tip speed begins to slow down. While this now older stripe is still alive, new streaks are created at a decreasing rate and cannot balance the streak detachment, which as we will discuss in more detail below, occurs at the upstream tip. Consequently, the stripe shrinks.

We would like to recall at this point that the tip instability is driven by the large-scale circulation around the stripe[53] and hence the

**Table 1 | Weibull fits to lifetime curves**

| | Re | $k$ | $\tau$ | Figure |
|---|---|---|---|---|
| Channel (experiment) | 600 | 5.9 | 1085 | 2(b) |
| Channel (experiment) | 620 | 9.5 | 1490 | 2(b) |
| Channel (experiment) | 630 | 8.1 | 2300 | 2(b) |
| Channel (DNS) (fully localized stripes) | 620 | 8.7 | 1055 | 2(c) |
| Pipe | 1740 | 1.3 | 55 | 2(b) |
| Pipe | 1860 | 1.1 | 445 | 2(b) |
| Pipe | 1880 | 1.2 | 900 | 2(b) |
| Channel (DNS) (non-localized stripes) | 730 | 1.1 | 385 | 2(c) |

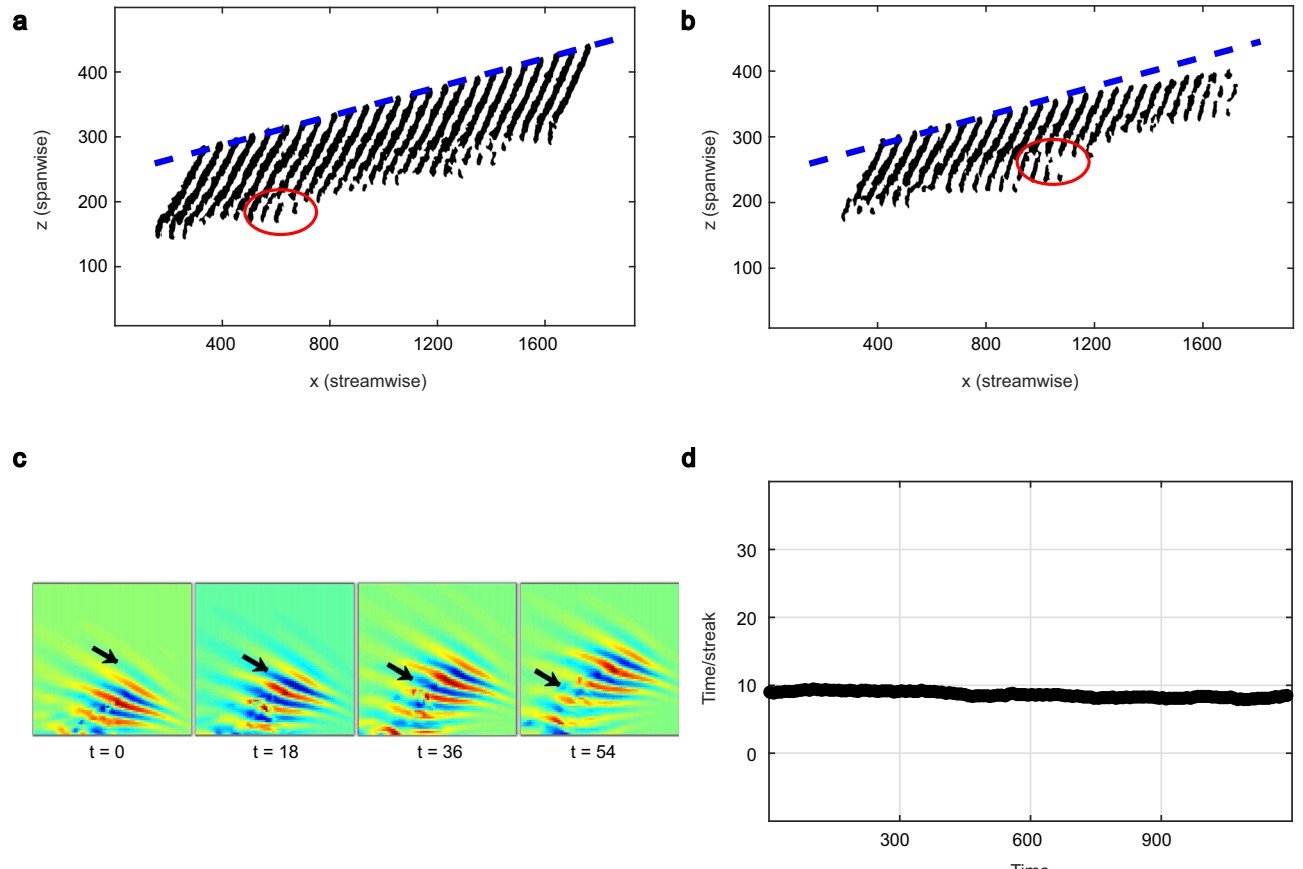

**Fig. 3 | Deterministic stripe expansion and streak creation. a** Consecutive positions of a stripe at Re = 655. The downstream tip moves at a constant velocity diagonally across the channel as indicated by the dashed blue line, and laminar fluid is entrained at a constant rate. **b** Below the critical point, at Re = 620, the tip initially moves with the same velocity as the stripe at Re = 655 but slows down in time and eventually the stripe decays. **c** Snapshots of the wall normal velocities in the region around the downstream tip of the stripe at Re = 655, obtained from DNS, showing a pair of streaks are created every ≈ 18 advective time units. To more easily follow the addition of streaks, the streaks emerging in the first snapshot ($t = 0$, arbitrary time origin) are shown with an arrow in all the subsequent snapshots. **d** The time interval between the addition of successive streaks shown as a function of time for Re = 655.

observation in experiments of a slowdown of the streak creation matches the observed slowdown of the large-scale flow; both these processes go hand in hand and comprise the aging aspect of stripes. While stripes shrink the interior turbulent motion, i.e., the existing streaks and vortices, remain unaffected by this process. This in turn also rationalizes the fundamental difference to other flows and specifically why fully localized stripes in Couette flow are memoryless. The latter do not rely on a tip instability to stay intact but instead new streaks are created throughout the stripe[55].

## Stripe fracturing and reproduction

In contrast to the deterministic and constant growth at the channel stripe's downstream tip, the dynamics at the tail (i.e., upstream tip) is stochastic. Stripe segments of varying size detach from the tail at irregular intervals. Some examples are marked by the red circles in Fig. 3a,b. The detached pieces typically dissipate. However, longer pieces occasionally survive and form a new stripe upstream and parallel to the parent stripe (see SI Fig. S2 for an example). To distinguish this process from stripe splittings in Couette[32] and small domain channel[40] simulations we will in the following refer to this process as stripe fracture. The key difference between the two processes is that during a split a gap opens along the stripe parallel direction, whereas for fractures the gap forms perpendicular to the stripe[55]. Consequently, splits depend on the stripe width, whereas fractures require stripes of sufficient length. In the vicinity of the critical point, channel stripes (unlike their Couette counterparts[55]) do not split according to

above definition but new stripes are exclusively created by stripe fracture. We first investigate the parent stripe's probability to fracture. For this analysis, we disregard the frequent detachment of small segments but focus on fractures larger than $20h$. As previously mentioned such large stripe segments occasionally grow and persist. For now our focus is solely on the parent stripe. Its fracture probability is determined irrespective of the further development of the daughter stripe.

The ability to fracture ultimately stems from the parent stripe's continuous extension, driven by the downstream tip instability. This constant growth rate between fracture events can be used to convert the stripe's length into a time, which may be interpreted as the intrinsic age of a stripe, where $t = 0$ corresponds to the extrapolated time where the stripe length is zero. The fracture statistics are shown in Fig. 4a, where the probability of a stripe to stay intact (i.e., not fracture) is shown as a function of length of the stripe. Stripes shorter than $200h$ have a fracture probability close to zero. However, as time proceeds and their lengths continue to increase, stripes appear to lose their structural stability and the probability to fracture increases. The faster than exponential decrease of the survivor function (Fig. 4a), with a value of $k = 5.7$ (see Table 2) which is obtained by fitting (Eq. (1)), attest that also the fracture probability of the parent stripe is not memoryless but an aging process.

We further illustrate the coupling between stripe extension and stripe fracture in Fig. 4b. Triggered close to the inlet, the stripe's length increases as it advects downstream. This growth is determined by the streak creation at the downstream tip and results in a close-to-constant

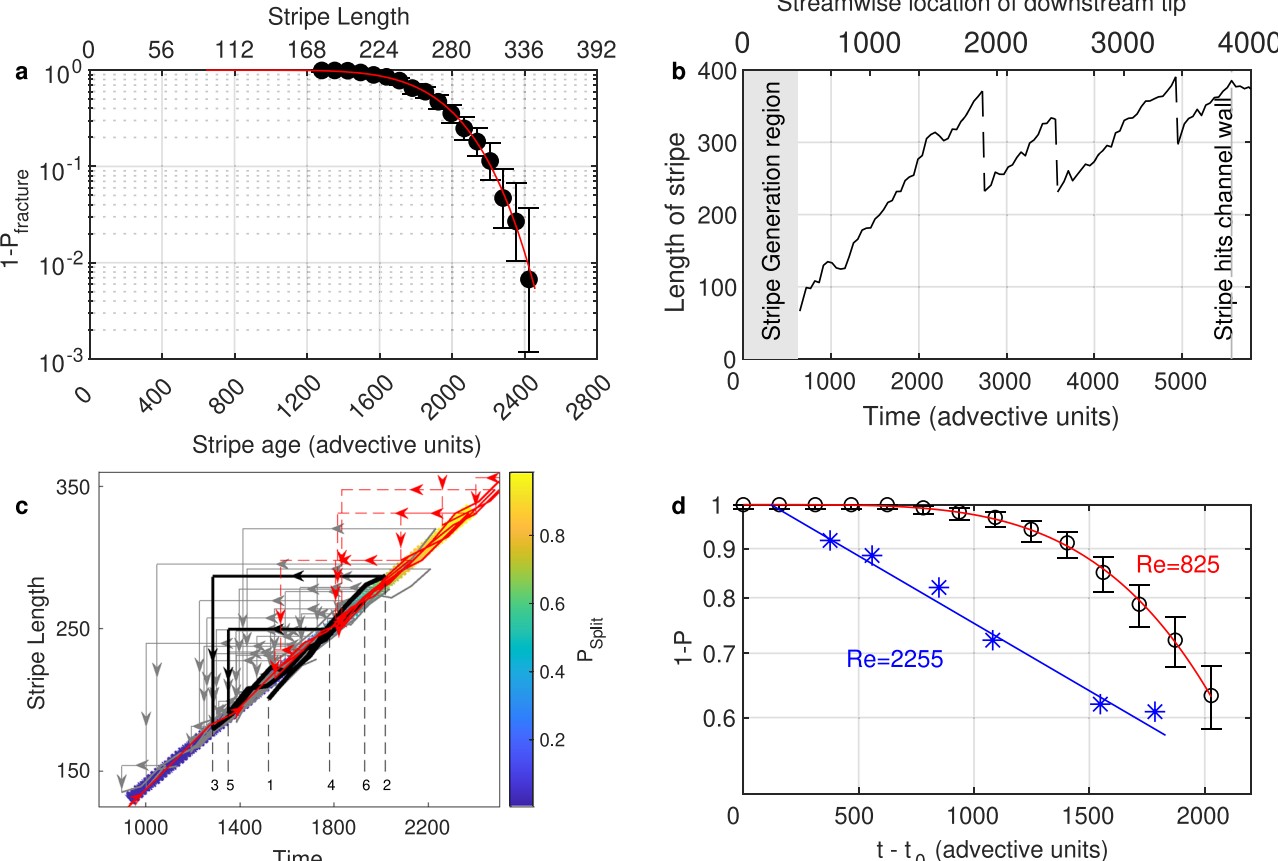

**Fig. 4 | Stripe fracturing and reproduction probabilities. a** Probability of a stripe to stay intact and hence not to fracture ($1 - P_{fracture}$). Error bars indicate 95% CI. Due to the constant rate of extension of individual stripes, the stripe length is proportional to time and stripe length can be interpreted as stripe's effective age. As shown, short (/young) stripes do not fracture and only once their length exceeds $\approx 200h$ the fracturing probability becomes nonzero. **b** An example of a stripe's length evolution (Re = 710) during its journey through the channel. At each fracturing event the original stripe is shortened and hence has a lower probability to fracture (see (**a**)). **c** Growth and fracturing sequences of a stripe. Since stripes grow linearly in time, the stripe's size is proportional to its effective age. In turn the probability to fracture increases with stripe length, and once fractured, the original stripe is shortened, which can be interpreted as the return to a younger effective age. As shown by the grey curves, localized stripes are subject to this cyclic growth-fracturing process as they move downstream. The diagonal line corresponds to the

growth of a stripe in the absence of fracturing, while the colour gradient illustrates the increasing probability for it to fracture. The horizontal and vertical arrows indicate fracturing events which reset a stripe's length and effective age. The black curve highlights the evolution of one stripe, for which numbers have been added to clarify the sequence of fracturing/reset events - see the main text for details. Note that here only fracturing of stripe segments larger than $20h$ are shown for clarity. To compare the dynamics of stripes in our experiment to those in earlier computational studies we analysed the data from Shimizu and Manneville[45] (supplemental movie at Re = 725) and the stripe length evolution for a representative time interval ($\approx 5000$ advective units), shown in red. **d** Reproduction probability, i.e., the fracturing probability, only counting events at which daughter stripes grow and survive. Error bars indicate 95% CI. The corresponding survival curve (red) is of type 1, aging, whereas the puff splitting curve shown for comparison[30] is of type 2, memoryless.

length increase. With length, the fracture probability increases (see Fig. 4a). Eventually, once the stripe approaches a length of $400h$, a substantial part of its tail detaches. While the original stripe is now shortened, the streak creation at the upstream tip is unaffected, and hence the stripe length increases again until the next fracturing event occurs. Consequently, the coupling between the growth of a stripe and its fracture probability leads to a cyclic process which can be visualized (Fig. 4c) by interpreting the stripe length as an effective age (assuming growth from zero length). The length evolution is depicted for several randomly chosen stripes (grey curves) at Re = 710. A representative stripe is highlighted in black. Starting from point 1, it initially grows at a fixed rate (diagonal line in Fig. 4c), and this slope is dictated by the streak creation rate at the downstream tip. With the stripe's length also its probability to fracture increases (shown by the color gradient) and in this case fracturing eventually occurs at point 2. This event resets this stripe to a shorter length (point 3) and hence a younger effective age, and the growth resumes until the next fracture occurs (point 4). The stripe is reset to point 5, and grows until point 6, at which point it exited the test section. Corresponding cyclic dynamics (light grey

curves) are exhibited by all stripes regardless of initial conditions. The interplay between stripe elongation and fracture causes the overall proliferation of turbulence to be two dimensional.

While most earlier studies were performed in domain sizes insufficient to capture such dynamics, recently two computational studies[45,47] could accommodate stripe lengths comparable to those observed in our experiments. Although neither the cyclic stripe dynamics nor the dependence of fracturing on the stripe length have been noted in either study, the available data allow to retrospectively test some of these aspects. The simulation at Re = 725 (supplemental movie in Shimizu and Manneville[45]) is sufficiently close to our measurements (Re = 700) to directly compare the stripe dynamics. In order to do so we determined the length evolution of turbulent stripes in this computation and added the data to Fig. 4c (red lines). Like in our experiments, stripes grow monotonically and they do so at the same rate, i.e., following the same diagonal. Visual inspection confirms that this deterministic growth, just like in the experiment, stems from the continuous creation of streaks at the downstream tip. Once stripes have reached a sufficient length their size abruptly reduces due to the

**Table 2 | Weibull fits to proliferation curves**

|  | Re | k | τ | Figure |
|---|---|---|---|---|
| Channel (fracturing) | 710 | 5.7 | 1355 | 4(a) |
| Channel (reproduction) | 825 | 4.2 | 2440 | 4(d) |
| Pipe (splitting) | 2255 | 1.0 | 3090 | 4(d) |

detachment of their upstream tail. At this Re such stripe fractures appear to occur at slightly larger stripe lengths compared to our experiments. Due to the periodic boundary conditions and limited domain size of this simulation, fractured stripe pieces cannot separate from the neighboring stripes and consequently interact, which leads to the subsequent decay of either the daughter or the parent stripe. Besides these computational domain restrictions, the dynamics of individual stripes are in excellent agreement with our experiments, and in both cases fractures occur at a similar rate.

To further test the agreement between experiments and simulations we determined the distribution of size changes exhibited by the computed stripes[45,47], specifically we sampled the data at intervals of 100 advective time units and recorded the length change encountered at each such step. As shown in SI Fig. S5 for all studies of channel stripes, experiments as well as simulations, the corresponding length distributions are distinctly asymmetric. The sharp peak at small positive values results from the monotonic expansion at the downstream tip. Conversely, at negative values the distribution is flat and extends to very large sizes, and these events correspond to the abrupt size drops caused by the fracture of the upstream tail. These distributions hence capture the cyclic dynamics of channel stripes, and this behavior is in stark contrast to size changes of structures with memoryless properties in other flows, such as puffs in pipe flow (SI Fig. S5d, for Re = 2150). The latter shows symmetric (with respect to zero) distributions as expected for structures fluctuating around an average size.

While we have so far considered the fracturing probability irrespective of the fate of the daughter stripe, we next determine the same probability but only considering those events where the daughter stripe survives, which we will refer to as stripe reproduction (SI Fig. S2). Since at Re = 710 such productive fracturing events are rare (of order 1%), we carried out measurements at Re = 825 where the rate of surviving daughter stripes is far higher. As shown in Fig. 4d, also in this case the survivor curve for $1 - P$ is not memoryless, hence also the stripe reproduction is an aging process, in this case with $k \approx 4$. For comparison, we plot the splitting probability $(1 - P_{split})$ for puffs in pipe flow[30], which is clearly memoryless (see Table 2 for the respective values of $k$ and $\tau$).

## Discussion
The key processes that determine the critical point and sustenance of turbulence in channel flow, the proliferation and decay of stripes, are not memoryless. Instead, stripe decay and proliferation probabilities change in time, the shape of the respective distributions marks them as type 1 survivor functions and hence aging processes. In both instances, the temporal change of stripe properties is connected to the deterministic growth at the stripes' downstream tip. This is in contrast to other shear flows such as pipe and Couette flow where decay and splitting processes are memoryless. With respect to the recently proposed analogy between the transition in shear flows and directed percolation, our findings show that the central line of argument, i.e., that active sites with Poissonian statistics can be identified with individual puffs[20,56] (or stripes[32,33]), does not hold for channel flow. However, formally, this analogy only concerns the microscopic dynamics, and discrepancies therefore do not automatically violate the DP conjecture. Yet, as has been demonstrated for simple models (e.g., Chantry et al.[36]), even changes much more subtle than those

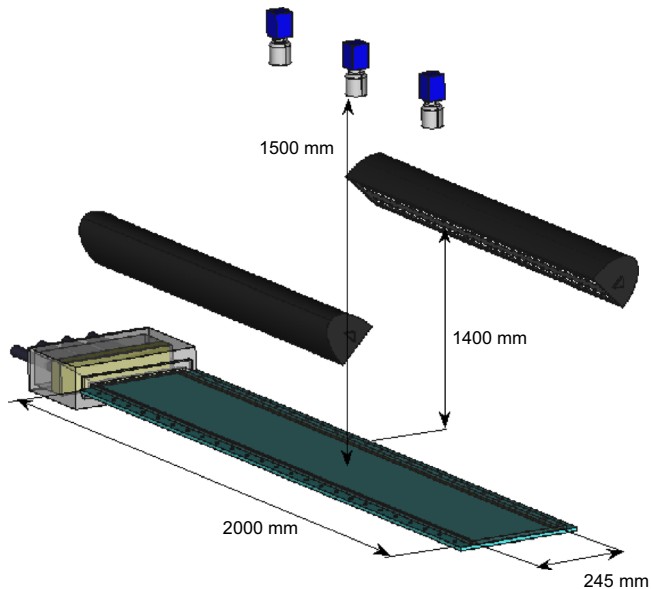

**Fig. 5 | Schematic of the experimental setup.** Prior to entering the channel, the water flows through a settling chamber (shown in grey) that smoothly converges from a height of 90 mm to the 1 mm (=2h) channel height. The channel (2000 mm in length and 245 mm in width) is illuminated via two LED lights (shown in black) and the flow visualization images (see text for details of the visualization technique) are recorded with cameras located along the channel.

uncovered here for channel flow can alter the universality class of the transition. To eventually determine if channel flow violates the DP conditions or if an analogy to DP different from that in other shear flows may exist would require the resolution of aspect ratios and dimensionless times far beyond those accessible in any shear flow experiment or simulation to date.

## Methods
### Experiments
**Experimental setup.** The experiments are carried out in a large aspect ratio channel which consists of two plates separated by a narrow gap of $2h = 1 \pm 0.03$ mm (Fig. 5). The bottom plate is a 10 mm thick, polished aluminium plate, while the top is 10 mm thick float glass. The gap is maintained by 1 mm thick steel strips, which are clamped between the glass and steel plates and form the side-walls of the setup. The size of the channel is $(L_x, L_y, L_z) = (4000h, 2h, 490h)$, where $L_x$, $L_y$ and $L_z$ are lengths in the streamwise, wall-normal and spanwise directions respectively. The working fluid is water, supplied from a continually overflowing reservoir, which is kept at the height of 21 m above the channel. The water level in the reservoir is maintained within $\pm 2$ cm, ensuring the pressure-head that drives the flow to be constant within $\pm 0.1\%$. The excess pressure drop due to an isolated stripe is less than 0.2% of the total pressure head and hence the flow rate remains constant during stripe decays or proliferation to within a fraction of a percent.

Prior to entering the channel, the water passes through a settling chamber including a porous barrier which breaks up large eddies. It then enters the channel proper via a convergence with an area ratio of 90:1. Unless perturbed, the flow in the channel remains laminar over the entire Reynolds number range investigated here.

The water temperature is measured just before the convergence by a precision resistance thermometer. The viscosity of the water $\nu$ is then determined using a fit to standard temperature-viscosity tables for water. The flow rate is measured by a magnetic flowmeter (ABB) installed in the main supply pipe leading to the channel, from which the bulk velocity $U_b$ is calculated.

To permit comparison with earlier studies, quantities are non-dimensionalized with the length scale h (half-gap) and 1.5 times the bulk velocity $U_b$, where $1.5U_b$ corresponds to the centreline velocity of a laminar flow with the same mass flow rate.

**Perturbation techniques.** To study the evolution of isolated turbulent stripes, an effective mechanism to generate them is needed. For Re > 900, almost any standard perturbation mechanism (static obstacle, moving obstacle, jets injected through one or more holes in the channel wall) can efficiently trigger turbulence. Although for the lower Re range studied here (600 < Re < 900), many of these do not work any more, two methods were found to be efficient in generating isolated stripes. In one, perturbing the flow over a certain minimum spanwise (≈20h) width successfully triggered stripe turbulence. In the present study, we realized this by placing a ferromagnetic obstacle 100h downstream of the inlet which can be actuated by impulsively moving an externally placed magnet across the upper channel surface. This extended perturbation generates a seed which evolves into a stripe.

In the other method, a stripe is generated at a Re higher than the target Re, where stripes are easily generated by a localized or extended perturbation (as described above). Next, the Reynolds number was quenched to the target value. This was accomplished with a bypass loop in the supply pipe leading to the channel, which was normally closed by a solenoid valve. Prior to triggering the stripe, the flow rate is adjusted to the target Re, with the bypass valve being closed. The valve is then opened, reducing the resistance in the supply pipes, increasing the flow rate and hence Re by around 100–150. After the stripe is triggered at this higher Re, it is allowed to develop for a time of around 450. Subsequently, the solenoid valve is closed, quenching the flow down to the target Re. Readings are taken after a further 375 to allow the flow to adjust to the change in Re, and for any transients to die out. In comparison to the quench, the generation of stripes by an extended perturbation becomes increasingly inefficient for the lowest Re studied here (Re < 700).

**Flow visualization.** The channel is illuminated with the help of LED lights installed parallel to the channel axis on both sides of the channel, ~1.4 m above it. The flow was visualized using reflective flakes 10–40µm in size (Eckart SYMIC C001 reflective mica particles). These flakes tend to orient along the shear and hence allows the distinction between turbulent and laminar regions. The flow structures are monitored in the test section with the help of three 4 MP cameras placed at a distance around 1.5 m above the channel. In order to allow time for the quench or the decay of initial transients, the test section is located some distance downstream of the channel entrance, and flows are recorded from 1500h downstream of the channel entrance till 500h before the channel exit. The combined view field of the three cameras is 2000h × 490h. To monitor the evolution of a stripe, images are captured by the cameras at a set frequency, with the three cameras being simultaneously triggered each time by a TTL signal.

**Image processing.** The images of the three cameras are merged to capture the flow field in the entire test section (Fig. 6a) and further processed by subtracting images of the laminar background flow (Fig. 6b). Images are next low-pass filtered using a Wiener filter with a 5 × 5 pixel mask. This is sharpened using unsharp masking in order to increase the contrast between the edges of the stripe and its surroundings (Fig. 6c). This can be further used for extracting quantities of interest. For instance, in order to determine the area occupied by turbulence, a simple threshold is subsequently applied to create a binary image, shown in Fig. 6d (in this example turbulence takes the form of a single stripe). The pixel count of the black area is then the area occupied by turbulence. For a reasonable range of parameters used in the image processing (e.g., the threshold for binarizing the image), there is only a minor variation in derived quantities such as the

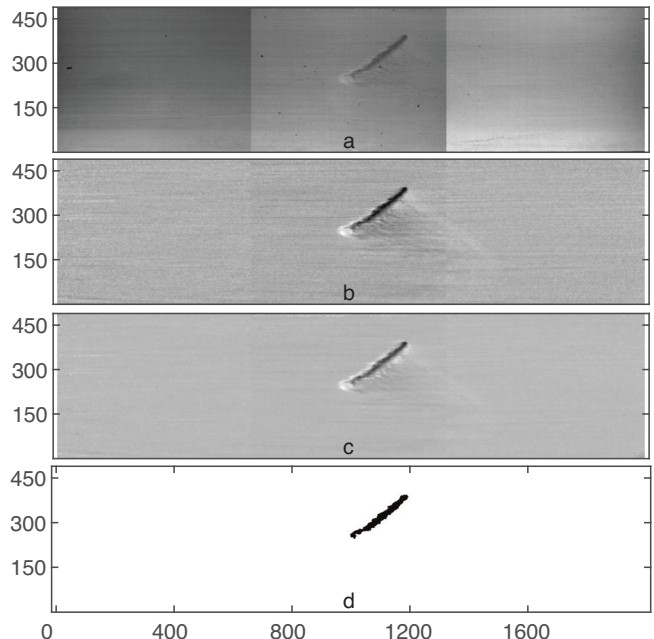

**Fig. 6 | Image processing. a** Raw image from the cameras stitched together showing an isolated stripe. **b** Background subtracted image. **c** Image after filtering and sharpening. **d** Binary image after thresholding.

area of the turbulence. More importantly, such changes do not affect the value of the critical point or the lifetime distributions.

The length of a stripe is determined using the binarized image. The image is divided into horizontal (i.e., streamwise) strips, each with a height of 5 pixels. For each of the strips that contains a part of the stripe, the centroid of that strip portion is determined. One then has a series of points that mark the stripe segment's center within each strip. The distance between adjacent points is determined and summed over to obtain the length of the stripe. Note that this is the actual length along the stripe which takes into account any bends or curves and it is not a projection onto a selected (inclined) orientation.

**Determination of growth rate and** $\mathrm{Re}_c$**.** The growth rate of turbulence was estimated as follows. For each Re, around 1000 measurements are carried out and in each case, the initial condition is a single stripe, typically generated at higher Re and quenched to the target Re. At a fixed time after the quench the length of the stripe is determined as described in the previous section. Next, we obtained the ensemble average over all measurements at that Re for the same time t (relative to the quench). This procedure was repeated for later times and in each case the total length of the stripe in the channel was determined, including shed stripe pieces and new stripes that may have originated from stripe fracturing. In doing so we obtain the evolution of the stripe length as a function of time. For each Re, this time variation can be approximated by a linear fit, whose slope is the growth rate at that Re.

The Re where the growth rate changes from negative to positive is suggested as an estimate (lower bound) of the critical point. Hence, similar in spirit to Avila et al.[30] we assume that the point where stripes begin to proliferate provides a close approximation of the actual critical point where turbulence first becomes sustained.

### Numerical simulations

The numerical simulations in the present study are carried out using a modified version of the hybrid (pseudo-spectral for the periodic directions and finite differences for the wall-normal direction) code `openpipeflow`[57], which was adapted to simulate channel flow[53] in a rectangular box.

$x', y', z'$ are the streamwise, wall-normal and spanwise directions, with the associated unit vectors being $\boldsymbol{e}'_x, \boldsymbol{e}'_y, \boldsymbol{e}'_z$ respectively. When scaled with half channel height $h$ and centreline velocity $U_{cl}$ of the parabolic profile with the same mass flux, the laminar base flow can be expressed as $\boldsymbol{U} = (1 - y^2)\boldsymbol{e}_{x'}$, and the Navier-Stokes equations for the fluctuating components $(\boldsymbol{u}, p)$ around base flow $\boldsymbol{U}$ take the form

$$\frac{\partial \boldsymbol{u}}{\partial t} + (\boldsymbol{u} \cdot \nabla)\boldsymbol{u} + (\boldsymbol{u} \cdot \nabla)\boldsymbol{U} + (\boldsymbol{U} \cdot \nabla)\boldsymbol{u}$$
$$= -\nabla p + \frac{1}{Re}\nabla^2\boldsymbol{U} + \frac{1}{Re}\nabla^2\boldsymbol{u} + \boldsymbol{f}(t), \nabla \cdot \boldsymbol{u} = 0. \quad (2)$$

Here, the Reynolds number is defined as $Re = U_{cl}h/\nu$, where $\nu$ is the kinematic viscosity of the fluid. $\boldsymbol{f}(t) = f(t)\boldsymbol{e}_{x'}$ is a time-dependent forcing term that represents an imposed streamwise pressure gradient. The mass flux in the streamwise direction $\boldsymbol{e}_{x'}$ is kept constant by changing the amplitude of the forcing term at every time-step.

No slip boundary conditions are imposed on the domain walls

$$\boldsymbol{u}(x, \pm 1, z) = 0. \quad (3)$$

**Computational domain.** The computations are carried out in a rectangular box shaped domain which can be tilted with respect to the streamwise direction[58,59] at an angle $\theta$, where $0° \leq \theta \leq 90°$.

The sides of the rectangular box $x$ and $z$ have associated unit vectors $\boldsymbol{e}_x$ and $\boldsymbol{e}_z$ respectively. The usual streamwise, wall-normal and spanwise directions have the following relation with the unit vectors associated with the domain directions.

$$\boldsymbol{e}_{x'} = \cos\theta\boldsymbol{e}_x + \sin\theta\boldsymbol{e}_z,$$
$$\boldsymbol{e}_{z'} = -\sin\theta\boldsymbol{e}_x + \cos\theta\boldsymbol{e}_z, \quad (4)$$
$$\boldsymbol{e}_{y'} = \boldsymbol{e}_y.$$

A constant mass flux is maintained in the streamwise direction and periodic boundary conditions are imposed on the faces normal to $\boldsymbol{e}_x$ and $\boldsymbol{e}_z$,

$$\boldsymbol{u}(x, y, z) = \boldsymbol{u}(x + L_x, y, z),$$
$$\boldsymbol{u}(x, y, z) = \boldsymbol{u}(x, y, z + L_z), \quad (5)$$

where $L_x$ and $L_z$ are lengths of the domain in the $x$ and $z$ directions, respectively.

The solver uses Fourier modes in the two periodic directions and finite-difference in the wall-normal direction. The velocity field is decomposed as

$$\boldsymbol{u}(x, y, z, t) = \sum_{k=-K}^{K} \sum_{m=-M}^{M} \hat{\boldsymbol{u}}_{k,m}(y, t)e^{i(\alpha k_x x + \beta m_z z)}, \quad (6)$$

where $k$ and $m$ are Fourier modes in the $x$ and $z$ directions respectively, $\alpha = 2\pi/L_x$, $\beta = 2\pi/L_z$. $L_x$, $L_z$ are the lengths of the domain in $x$ and $z$ directions, respectively. The time integration is performed using a second-order backward differentiation for linear terms and the Adam–Bashforth method for the nonlinear terms.

The evolution of the stripes is monitored by defining a scalar observable - perturbation kinetic energy $E(t)$ defined as

$$E(t) = \frac{1}{L_xL_yL_z}\int_{-1}^{1}\left(\int_0^{L_z}\int_0^{L_x}|\boldsymbol{u}|^2\mathrm{d}x\mathrm{d}z\right)\mathrm{d}y. \quad (7)$$

**Large domain with $\theta$ = 0°.** A fully localized stripe can be simulated only in a large enough domain. We studied the lifetimes of the stripes at $Re = 620$ in a square domain with no tilt, i.e., $\theta = 0°$ with size $(L_x, L_y, L_z) = (400, 2, 400)$ and resolution $(N_x, N_y, N_z) = (1536, 64, 1536)$.

The resolution used is sufficient for the simulations at $Re = 620$. The initial conditions for the lifetime study are uncorrelated snapshots from simulations at $Re = 700$ in the same domain. The lifetimes are

determined by setting an appropriate cutoff on the perturbation kinetic energy. Once the perturbation kinetic energy falls below this threshold, the stripe is considered to have decayed. However, unlike the case of the partially localized stripes, there is no plateau in the energy before the decay. Rather, the kinetic energy decays gradually (Fig. 2d). Hence, the lifetimes depend significantly on the threshold. The lifetimes for different threshold values are shown in SI Fig. S4. Though the lifetimes do depend on the threshold, the qualitative behavior remains the same, exhibiting a non-exponential decay of survival probabilities.

**Tilted domain.** A partially localized stripe, i.e., a stripe localized only in one direction and periodic in other direction, is simulated in a rectangular domain tilted with respect to the streamwise direction at an angle of $\theta = 45°$. This tilt angle also corresponds to the stripe angle with respect to the streamwise direction[60] and the angle of tilt chosen here agrees with the tilt angle of the stripes observed in the experiments and reported in Tao et al.[43].

The domain size is $(L_x, L_y, L_z) = (30, 2, 100)$ with resolution $(N_x, N_y, N_z) = (256, 49, 768)$. The initial conditions for the lifetime study are taken from uncorrelated snapshots from simulations at $Re = 600$. The stripe is considered to have decayed when the energy of fluctuations drops below a selected cutoff value, e.g., $E = 0.01$. Due to the rather clear plateau in energy before decay, the results do not strongly depend on the cutoff chosen (see Fig. 2).

## Data availability
Source data are provided with this paper.

## Code availability
The numerical simulations were carried out using the open source code `openpipeflow`[57].

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

## Acknowledgements

This work was supported by a grant from the Simons Foundation (662960, BH). We thank Yohann Duguet for helpful discussions, Baofang Song for the initial adaptation of `openpipeflow`[57] to the channel geometry, and Ashley P. Willis for `openpipeflow`[57].

## Author contributions

B.H. conceived the project. V.M. C.S.P., and M.P.S. carried out the experiments. C.S.P. carried out the channel simulations in the large domain. G.Y. carried out simulations of pipe flow as well as channel flow in the tilted domain. V.M. analyzed the data from the experiments and

simulations. V.M. and G.Y. contributed to the analysis methodology. B.H. wrote the manuscript with contributions from all the authors.

## Competing interests

The authors declare no competing interests.
