## [Transparent Peer Review file · Nature Communications]

Aging and memory of transitional turbulence

Corresponding Author: Professor Björn Hof

Version 0:

Reviewer comments:

Reviewer #1

(Remarks to the Author)

Plane channel flow is one of the most important examples of a wall-bounded shear flow and route to turbulence in this flow is currently of much interest. The key results in this study demonstrate that turbulent transition in this flow follows a different scenario than that recently established for other wall-bounded shear flows. These results show that neither decay nor splitting processes are memoryless for this flow. This is a significant contribution to the field. The nature of the splitting shown in figure 4 is particularly striking. The results reported are from state-of-the-art experiments of unprecedented size together with numerical simulations and the conclusions are strongly by the data presented. The methodology is sound and sufficient detail is provided in the supplementary material for the results to be reproduced.

I have three main (related) concerns that the authors need to address.

1) Figure 2(b) shows clearly the situation, contrasting metastable decay (typical of other shear flow) with the more-or-less monotonic decay observed in unconstrained channel flow. The latter is clearly not memoryless, but in what sense is the turbulence aging? Combined with what is shown in figure 1, I would say that the turbulence is contracting spatially. Presumably if one looks in the center of any of the turbulent patches the local energy, streak-roll dynamics etc are approximately constant (or as constant as the fluctuating energy shown in grey in the figure). There is no evidence that the internal dynamics of the turbulence is aging, and I would be surprised if this were the case. I would like the authors to consider whether aging is really an appropriate description to use in the title of the paper. If so, then I would like to see some clearer justification for this term.

To be clear, from the abstract I see no issues with: "We here show that for the classic case of channel flow, the growth of turbulent stripes is deterministic and that memory-lessness breaks down. Consequently, mapping between turbulent patches and active DP-sites is not fulfilled."

2) From the supplemental material: "Each stripe is considered at a definite time after the perturbation, its area determined and then an average found over all the stripes at that Re at the same time, thus yielding an ensemble average of the turbulent fraction at that time." I have concerns about this method of aligning the different samples. It is not clear that a fixed time after initiation will align the stripes to equivalent states. Why not align them based on a fixed value of the turbulence fraction or stripe length and look at the time to decay from that time? One would need to investigate the dependence on the value used to align the samples etc. One can already see in the DNS data for energy in Figure 2(b) that there may be some alignment issues. I would expect that using turbulent fraction rather than energy would smooth those decay plots.

3) A final point is related to the previous points. Consider the following. If the contraction of stripes occurred at exactly the same rate for all realizations and all stripes are aligned suitably to start with, then P_{survival} would be a step function, 1 until some common decay time and then 0 after. Now, due to intrinsic fluctuations of turbulence, and due to differences in initial conditions (and possibly issues with the alignment noted in 2), the decay times will not all be the same. Hence the P_{survival} will now be a rounded step function. The question is, are we seeing something fundamentally different from this? Is there a signal in the lifetimes that is different from just broadening due to the factors just listed? Again, is this "aging" in that

there is some significance/universality in the form of the survival functions, or is it instead a broadened deterministic decay process? I appreciate that this might be difficult to address and so I do not necessarily expect the authors to fully address this in the revised manuscript, but I will ask them to consider the question as it is clearly important.

Minor comments:

"Note that in all the experiments, stripes remained at a minimum distance of $50h$ from the lateral boundaries". I understand but suggest instead: "Note that in all the experiments, stripes remained at least a distance of $50h$ from the lateral boundaries".

Figure 2(a): the authors should comment on why $Re=620$ from DNS is like $Re=600$ from experiments, or at least alert the reader. I was confused.

Figure 3(c): maybe some faint lines separating the panels.

"Aging dominates stripe dynamics below as well as above the critical point and both aspects are closely connected to the deterministic growth at the stripes' downstream tip, an aspect that is crucially missing in many previous studies of channel flow." Apart from the issue of the term aging, this is misleading in that there are several recent studies on the growth of stripes, and it is widely recognized that this will affect the nature of transition.

Figures 2 and 3 in the supplementary material: captions refer to (a), (b), (c) but there are no labels in the figures.

Reviewer #2

(Remarks to the Author)

This paper investigates the time evolution of turbulent stripes around the critical point to turbulence in channel flow, and reports that the evolution is far from memoryless, the Poisson process. For this reason, the authors conclude that the theory of directed percolation (DP), which requires the Poisson process, is invalid to channel flow while many previous studies have shown the adaptability of the theory.

The experiments and numerical simulations in this paper are very huge and valuable, and I imagine the authors carefully analyzed the big data. Especially, Figure 4b is a very meaningful diagram clearly showing the characteristics of a turbulent stripe.

Although the presentation of experimental data is very good, some of their interpretations are fatally lacking in evidence. If the comments below are not misguided, I cannot recommend this paper to Nature communications.

(1) The authors investigated only growth along turbulent stripes. However, there are several kinds of growth/decay processes in channel flow. Furthermore, the splitting rate at a given length (P_{split} in Fig.4b) may be memoryless. Since a rich variety of stochastic models exhibits DP behavior, they must show more clearly why the deterministic growth spoils the requirements of DP theory.

(2) Although their experimental equipment is very huge, it is not still enough to obtain equilibrium single stripes. The growth rate in Fig.1 makes sense only for specific initial conditions. Since the turbulent intensity along a stripe is not uniform, it has a finite equilibrium length which depends on Re .

(3) The word "aging" seems reasonless to me. Since $Re=620$ is too far from the critical Re , the decaying trajectories in Fig.2 are inevitable. Previous studies observed DP behaviors only within the small range of Re . It is necessary to investigate decay/growth in the vicinity of the critical point to show the difference from other flows. Although the motion at the downstream edge of a stripe is close to periodic, a large deviation may abruptly turn it to decay. I have seen the numerical result of a single strip which decays after long transient $O(10^5)$.

Reviewer #3

(Remarks to the Author)

The manuscript of Mukund et al. describes exhaustive experiments in a very long channel (by respect to the height) where a liquid flows with a Poiseuille profile. They studied the transition to turbulence in this facility following the life of spatial-temporal structures in the form of isolated localized bands or stripes, build with individual short streaks.

The authors claim here that, at difference with previous similar experiments and numerical simulations, they observe that the

lifetime distributions or survival probability of structures at Reynolds number below the critical one, does not show exponential distribution as it is required for memoryless process.

The experiment is very well conducted in this unique facility which allows to study, by fluid visualization, very long trajectories of the stripes in the channel and then to build very important statistics. A accompanying numerical simulation is performed with the code `openpipeflow` which has been used before in well recognized studies on transition.

The flow was visualized using light reflecting tracers whose inclination and the light intensity reflected depend on the velocity gradient tensor, but which is not capable to discriminate the components of the small-scale flow (SSF) by respect to the large-scale flow (LSF). It is indeed known that this distinction is very important in transition to understand the evolution and dynamics of turbulent spots and other structures.

Klotz (2021), using Particle Image Velocimetry techniques (PIV), observes and shows in this Figure 20 an oblique LSF with orientation parallel to the turbulent band reaching the highest intensity near to the laminar–turbulent interface. From DNS simulations, Gomé (reference [39] in the manuscript) refers to the importance of this LSF during laminarization. Couliou (2017) shows the role it plays in the creation of streaks. This incompressible large-scale circulation is sensitive to the position of the side walls of the channel, so its influence may be different in wide or straight channels.

So, it is fundamental to verify if the existence of this more homogeneous circulation, the LSF, that could be at the origin of many description made by the authors on the evolution of the stripes. It is the case, for example, to understand what is discussed in the Figure 3 and the long persistence of the stripes before the abrupt decay. It is very important because these elements could lead to the main conclusion of the study that is to say the absence of memoryless process.

As the flow visualization techniques used cannot discriminate scales, one immediate and reasonable possibility for confirming or for disproving these conclusions is to revisit the existing data obtained in the accompanying DNS, filtering long and small scales.

In addition, a qualitative view of the LSF could be obtained, as in the pioneering experiments of LSF in Rayleigh–Bénard convection, in a new run of the experiment using dyes or following the lagrangian trajectory of particles.

Another key question about the validity of the results exposed in the manuscript arises from another limitation of the flow visualization technique, which is not capable to distinguish between decay times for the different small-scale velocity components that is to say streaks and rolls as it has been put in evidence recently by Liu (2021).

These results need to be taken into consideration as the LSF can survive longer times, perhaps by viscous mechanisms (see Rolland (2015)). The LSF may distorts the scenario of the aging of stripes.

Once again, the possibility to revisit the existent numerical data of the DNS can be put light around these open questions.

The paper is well written. From this point of view, only few paragraphs deserve to be modified to gain clarity. In particular the last one in page 8 which refers to the influence of geometrical changes or imperfections which is not comprehensible for this referee. Is it related to the influence of noise or external perturbation as already discussed in Liu (2021)?

Minor points

p.2 :

Cut out the irrelevant reference to the bitcoin blocks as memoryless process, which is questioned by Bowden, R., Keeler, H. P., Krzesinski, A. E., & Taylor, P. G. (2018). Block arrivals in the Bitcoin blockchain. arXiv preprint arXiv:1801.07447.

p.3 :

Begin the references of DP and turbulence [31-36] with Pomeau (1986) who first suggested that the propagation of laminar turbulent fronts may be determined by the critical exponents of directed percolation.

It is said that "In the transitional regime, turbulence appears in the form of stripes", but it is necessary to briefly discuss the fact that in PPF, Carlson (1982) and Lemoult (2014) have observed spots and that Tsukahara (2014) has observed bands.

p.4

In the assertion "1.5 times the bulk velocity u_b (corresponding to the centre-line velocity in case of laminar flow)" a confusion may arise with the text in parentheses which could be erroneously attributed to u_b .

p.4 and p.1 in supplementary information:

As the initial perturbations to generate strips are obtained moving the ferromagnetic obstacle, how is the influence of his inclination motion across the channel?

p. 5, Fig. 1 and p.3 and 4 in supplementary information:

How is exactly measured the growth rate? It is necessary to show a figure with the data leading to this estimation (time evolution of the turbulent fraction). Is this quantity the inverse of a typical life time? If it is the case, these are not very long (only 10 units for $Re = 620$)

There are only 3 values of the growth rate before the extrapolated value of Rec . What are the criteria for choosing 4 values (and not 3 or 5, even 6)? Has the critical value be obtained by extrapolation of negative growth rate or by change of sign?

p. 5: It is mentioned in the manuscript that the value of $Rc = 650$ agrees with references [41-44], which seems to give higher

value. How to compare with the equivalent DNS study in the reference [39]?

p. 5 and Fig. 2:

In the reference [39] it is observed that a turbulent band persists as a long-lived metastable state before decaying abruptly. In consequence, his Figure 9 presents the evolution of spectral quantities and velocity field norms in function of time plotted with a translated time beginning when decay and exponential decay are observed. What happens if this procedure is also applied on the results of this work?

p. 13, Fig. 3a and 3b:

The spanwise direction is Z and not Y as is noted in the figures.

REFERENCES TO BE INCLUDED:

Carlson, D. R., Widnall, S. E., & Peeters, M. F. (1982). A flow-visualization study of transition in plane Poiseuille flow. *Journal of Fluid Mechanics*, 121, 487-505.

Couliou, M., & Monchaux, R. (2017). Growth dynamics of turbulent spots in plane Couette flow. *Journal of Fluid Mechanics*, 819, 1-20.

Klotz, L., Pavlenko, A. M., & Wesfreid, J. E. (2021). Experimental measurements in plane Couette–Poiseuille flow: dynamics of the large-and small-scale flow. *Journal of Fluid Mechanics*, 912

Lemoult, G., Gumowski, K., Aider, J. L., & Wesfreid, J. E. (2014). Turbulent spots in channel flow: an experimental study. *The European Physical Journal E*, 37(4), 1-11.

Liu, T., Semin, B., Klotz, L., Godoy-Diana, R., Wesfreid, J. E., & Mullin, T. (2021). Decay of streaks and rolls in plane Couette–Poiseuille flow. *Journal of Fluid Mechanics*, 915.

Pomeau, Y. (1986). Front motion, metastability and subcritical bifurcations in hydrodynamics. *Physica D: Nonlinear Phenomena*, 23(1-3), 3-11.

Rolland, J. (2015). Mechanical and statistical study of the laminar hole formation in transitional plane Couette flow. *The European Physical Journal B*, 88(3), 1-10.

T. Tsukahara, K. Iwamoto, H. Kawamura, and T. Takeda, "DNS of heat transfer in a transitional channel flow accompanied by a turbulent puff-like structure," preprint arXiv:1406.0586 [physics.flu-dyn] (2014).

Version 1:

Reviewer comments:

Reviewer #1

(Remarks to the Author)

The authors have conducted additional experiments and have addressed the concerns of my original referee report. The results in the paper are significant and the message is important: Memoryless decay is not inevitable for turbulent structures in wall-bounded shear flow. For some time, the onset of turbulence as a continuous DP transition was not clearly understood/accepted within the scientific community, and then more recently this has given way to an equal misconception that all subcritical transitions to turbulence are of DP type. As the authors correctly point out, the simple model in Ref. 36 illustrates that the issue is in fact subtle. Here the authors investigate extensively the characteristics of stripes in channel flow and show that memoryless decay is not in operation.

My view is that the work meets the standards expected for publication in Nature Communications. However, I cannot quite recommend publication in the current form. There is one significant remaining issue that I will ask the authors to address. (I may have missed in the first review, or it may have arisen from the reworking of the paper in the revision process).

As far as I know, there is no equivalent study of the decay process of unconfined stripes in any other shear flow unconstrained in two dimensions, e.g. plane Couette flow or unconstrained circular Couette flow. Hence, it is not clear that the nature of stripe decay presented in the manuscript results from considering channel flow, or from considering unconfined stripes. I have no doubt that the downstream growth mechanism of stripes in channel flow leads to fundamental differences with Couette flows. However, in the decay regime, the differences between the flows may be less important.

This is obviously a big question, potentially at least, but I want to the author to consider it. At a minimum they need to address this possibility and to clarify that all past studies of memoryless decay are for flow confined in all but one direction, or if they know of a memoryless decay in a flow unconfined in two directions, they can state that. Another option is for the authors to consider data from circular Couette flow (which they should have access to) and to extract decay statistics for

stripes in that system. That would be considerable work. This issue is sufficiently important that I will insist that the authors address it, but I will leave it for them to decide how.

Reviewer #2

(Remarks to the Author)

The author responded to my comments point by point, but I still have doubts about the main claim of "deterministic growth of stripes".

The size of isolated stripes, like turbulence in a pipe flow, fluctuates around a Re-dependent average. This is already a well-known fact. It has been reported in several papers that a stripe survives at least about 10^4 h/U at Re=700 while keeping its average length. In this sense, the authors' experimental domain size, which corresponds to an advection time of 1000 h/U, is still too small to consider their main claim. Figures 1 and 2 may only represent the initial transient period due to the perturbations. As to the distance from the critical point, the comparison with the pipe flow is not meaningful. To clarify their main claim whether stripes are memoryless or not they have to investigate stripes for $650 \leq \text{Re} \leq 700$ using much larger domain.

For the reasons stated above, their results are not well-established and may be misleading to readers in other fields. We do not believe that this paper should be published in this journal.

Reviewer #3

(Remarks to the Author)

The new version of the manuscript, which integrates my comments, deserves to be published.

Version 2:

Reviewer comments:

Reviewer #1

(Remarks to the Author)

The authors have fully addressed the question in my previous report and I recommend acceptance.

Reviewer #2

(Remarks to the Author)

The biggest point of controversy between the authors and I is whether a turbulent band in channel flow is "aging" or "fluctuating". See the Figure 5.1 in Kanazawa's PhD thesis (https://ir.library.osaka-u.ac.jp/repo/ouka/all/69614/29805_Dissertation.pdf), where bands at around Re=700 ($440 < \text{Re} < 480$) persist for a long time, and they can define an average length at each Re. Also see the animation of the PRL supplement at Re=725 (<https://journals.aps.org/prfluids/abstract/10.1103/PhysRevFluids.4.113903>) which shows that band persist at least for a few 10^4 (H/U). There are a few 10^3 (H/U) transient phase before a band becomes the fluctuating state. The paper by Xiong, the authors mentioned, is a bit old and their domain size is too small to investigate band's length. Now several groups numerically show sustained single bands in large domain. A single band at Re=700 in a very large domain can survive longer than 10^5 (H/U) without splitting and decaying. No matter how large the experimental setup is, it is difficult to achieve the "fluctuating" state in the present experiment.

For these reasons, I again cannot agree with the author's idea, "aging" of bands. Although this paper contains interesting and meaningful experiment, it gives the wrong impression to readers who are not familiar to this subject. A more specific journal is suitable to this paper.

Version 3:

Reviewer comments:

Reviewer #2

(Remarks to the Author)

As for the biggest claim, "memory of bands," I still disagree. First, regarding the survival probability in Fig.2, it is obvious that bands decay monotonically at around Re=600~630. This is because the downstream end cannot be maintained for this range of Re. Much longer time scale $O(10^4-10^5)$ experiments around Re=700 are needed to determine memory or memoryless. In this sense, it is impossible to answer this question with the size of the authors' experimental apparatus.

The authors also associate memory with the non-exponential tail in Fig.4a, but this is meaningless. Fig. 4b is a similar to the length time series in Kanazawa's paper that I sent in my previous comment, but it does not necessarily show "successful splitting" when a band becomes shorter. For example, a single band is maintained for a long time period $O(10^4)$ at Re~700 during repeated stretching and shrinking. A successful split is when a new band is formed stably next to the old one. "The success or failure of a split is memoryless." This is also the case for pipe flow (Re~2200). A puff grows in length and splits by

chipping, but the downstream one does not necessarily survive as a new puff. If it disappears immediately, we do not call it splitting.

It is well understood that a single band repeats stretching and shrinking in time, but in order to examine whether decay or splitting is memoryless or not, it is necessary to observe for much longer time $O(10^4-10^5)$ for $Re=660-700$. I am sorry to disagree with you again and again, but I cannot accept the authors' claim, memory, based on the current experimental results.

Reviewer #4

(Remarks to the Author)

Review of Memory of transitional turbulent structures

General Comments

The present reviewer has been asked the following: "You will see that this manuscript has been reviewed by three referees in the past. We are now in a bit of difficult situation – reviewer #1 cannot continue to review the revised version in response to reviewer #2's remaining concerns. We are keen to ensure that the original concerns raised by reviewer #2 are scrutinized by another expert like you. We would thus greatly appreciate if you could help us to check the authors' point-by-point response. Of course, please don't hesitate to let us know if there were any other places that you would like to draw our attention."

In the response to reviewer 2, it is written, "The biggest point of controversy" is if stripes are "aging" or "fluctuating". We would not have phrased it quite this way. The biggest point is if stripes are memoryless or not. While a decrease in life expectancy with time (our Fig. 2) is the literal meaning of aging, we appreciate that this term may carry other implications that are not intended by us. To avoid misunderstandings we have decided to remove the term "aging" from the manuscript and we directly refer to the time dependence of stripe properties and we hence keep the focus on our core message: the lack of memorylessness and the implications this fundamental change has for the transition to turbulence."

This reviewer sees two major issues:

1. The first is language. The second reviewer and the authors' disagree primarily because language that defines and describes physical phenomena is not precise. It is used in a qualitative way. It is suggested that this can be overcome through modification of the paper. For instance, for flow physics describing aging, could be defined in a quantitative and not qualitative way near the beginning of the paper. Suggest equation(s) or mathematical properties that can be defined and applied to both the DNS and experimental measurements. For example, memorylessness should have a quantitative definition in the paper (and not referenced away) so that readers will have a firm understanding of what it is, how it is quantified, and in this case why the authors' think it is not present for the particular flow-field. This way, the ambiguity of the definition and a lack of quantification of these terms can be overcome.

2. Second, comes from my own review of the paper. Multiple points regarding what are new are buried within large sections of text (without paragraph breaks). It is suggested that in the introduction of the paper that the authors' are very a) clear, b) up front, c) explicit, and d) quantitative about the claims they are making in the paper. These statements are present, but the reviewer had to work hard to find them.

Otherwise, the points back and forth between reviewer 2 and the authors' are reasonable, and the reviewer believes comes from the fact that reviewer 2 and the authors' essentially have different viewpoints or definitions of qualitative wording to describe a quantitative subject. In science and math, it is advisable to be as quantitative and explicit with definitions as possible. This would avoid this entire situation. Then the paper can be assessed with the same basis and understanding.

By making these two changes, the reviewer believes that the disagreements between authors and previous reviewers can be resolved.

Specific comments on Writing in Revised Article

For a manuscript of this length, there is an overly lengthy introduction through pages 1 through half of 4. The reviewer has no idea why all this introduction material is important until the author states what they are doing midway through page 4. It would be better to inform the reader what the intention of the paper is in the beginning. Also see general comment 2. Major implications of the paper are spread too far apart and are buried in particular paragraphs.

Page 5 – Awkward to have a single sentence as a paragraph.

Page 5 – Spacing between turbulence , - e.g. no space.

Page 5- spacing between (and stripe

Page 5 – Should have paragraph break near end before We propose... Suggest moving this core idea earlier as it is buried

within paper and an important point.

Page 7 – Buried comment regarding research performed regarding DNS that was not mentioned explicitly prior.

Page 8 to 10 – Please use paragraph breaks in your writing to separate core ideas. This is fundamental in technical writing. It makes it difficult to understand core ideas in your writing.

Noting early on that some spacing is not made between references and words.

SI Page 6 last line – undefined reference with ? .

Note: It would be appreciated to have line numbers on manuscripts to review for easier reference.

Version 4:

Reviewer comments:

Reviewer #5

(Remarks to the Author)

In this paper, the turbulent stripes in channel flows are studied experimentally and numerically, and neither their decay nor their proliferation are found to be memoryless. It is concluded that the DP analogy and the chaotic saddle feature of turbulence do not apply to individual localized stripes. It is hard to produce and study the turbulent stripes in experiments at low Reynolds numbers, and the results are important to understand the characteristic turbulent structure and are helpful to evaluate the present physical models. Therefore, I'd like to recommend its publication after the following concerns are addressed.

1. Stripe length is a key parameter for this study, but is not defined in both the manuscript and the supplementary file, and should be added in the new version. Is it the length in the oblique direction? the streamwise direction? or the spanwise direction? How is it determined in the image processing and in the numerical simulations?

2. It has been shown that the DP analogy is applicable for the stripes in channel flows as Re is larger than 1000 (e.g. Phys. Rev. Fluids, 113902, 2019), so the statement in the abstract “the one to one mapping between turbulent stripes and active DP-sites is not fulfilled” seems arbitrary.

Different from most individual stripes as $Re > 1000$, the stripes at low Re are isolated turbulent ones: owing a clear downstream head and an upstream tail, and this is the reason why the flow state at the low Reynolds number range is referred to as the sparse turbulence, where the DP analogy is not applicable (Phys. Rev. Fluids, 011902, 2018). Therefore, it is suggested to emphasize that the analyses and statements are made for the isolated turbulent stripes at low Reynolds numbers or at the sparse turbulent state.

3. The initial conditions, e.g. the initial kinetic energy in DNS or the initial length of the stripe, have substantial effect on the lifetime and the growth rate of stripes: longer initial stripes may have larger growth rate in a longer period before their final decay (see Phys. Fluids, 041702, 2015). Consequently, some turbulent stripes may leave the outlet of the channel in experiments before starting their decay process and are identified as growth cases, leading to an underestimation of the critical Reynolds number. We understand that the experimental channel cannot be infinitely long, but it is suggested to add figures, showing the length (or kinetic energy) distribution of the initial stripes, and the corresponding discussions.

4. The second key point in the abstract is about the chaotic saddle concept, but this concept is not mentioned at all in the manuscript and the supplementary file except the introduction part, and hence it is suggested to add some words explaining the relation between the memoryless decay and the chaotic saddle.

Minor questions

a) Line 113-117: “Another recent numerical study...”, the corresponding reference should be added.

b) Line 241: “...by the upstream tip...” should be “by the downstream tip”?

Reviewer #6

(Remarks to the Author)

Dear Editor,

I have read the article and supplied reviews. I am satisfied that the points raised have been addressed, details follow. I believe that the methodology is sound and the results are noteworthy. I have only a couple of minor comments of my own, I do not require to see another draft.

Reviewer #2

1)

On this, the key point (regarding aging vs a memoryless process), the authors have provided a robust response to referee. The characteristic properties of aging are (now) described mathematically and are supported by data ('new figure 2b'). (From the response to Reviewer #4, it appears that much of this was missing in the previous draft.)

2)

The reviewer states a decay property of the bands. The authors present this in figure 2d(upper). I agree with the authors that the decay property does not equate to a trivial (viscous) decay process -- this is the 'aging' that adjusts the decay rates of the superimposed stochastic process.

3)

The referee requests longer times to see if the process is memoryless. While the author response is long, it can be quickly seen that a memoryless would require a stationary nature of the bands, contradicting the referee's own observation in point 2) about the slow decay of the bands.

4)

This is a detailed matter on the nature of the stability between different flows. I think there is room for differing interpretations. The authors present a reasonable argument and have added text on this.

5)

Like 3), the referee's assertion that the success or failure of a split is memoryless seems unlikely, as the background decay of the bands will affect this process.

Reviewer #4

1. The requested mathematical framework has been added.
- 2, 4. Presentation of the work appears to have been addressed.

My own comments:

In the abstract "The discrepancy between channel flow and the established transition model..."

It would be more precise and clearer to say more explicitly, "... between channel flow and the transition models established for Couette flow, pipe flow..."

Line 177, please clarify, including fully localised stripes in Couette flow?

Version 5:

Reviewer comments:

Reviewer #5

(Remarks to the Author)

The authors have answered my concerns, and I am satisfied with the revised version and recommend it for publication.

Reviewer #1:

We thank the referee for the helpful comments. Prior to answering the specific questions, we would like to mention that in the revised manuscript we include additional lifetime data (Fig. 2a). This data has been recorded substantially closer to the critical point (as requested by another referee) and it confirms our previous observation. The initial phase during which no stripes decay is further prolonged and when stripe decays eventually set in the process is not memoryless. Hence the probability to decay/die increases in time, which is what we refer to as stripe aging. To obtain this new data required a complete reassembly and modification of the experiment and this was only completed recently, and caused the long delay in re-submission.

Regarding the specific questions:

1) Figure 2(b) shows clearly the situation, contrasting metastable decay (typical of other shear flow) with the more-or-less monotonic decay observed in unconstrained channel flow. The latter is clearly not memoryless, but in what sense is the turbulence aging? Combined with what is shown in figure 1, I would say that the turbulence is contracting spatially. Presumably if one looks in the center of any of the turbulent patches the local energy, streak-roll dynamics etc are approximately constant (or as constant as the fluctuating energy shown in grey in the figure). There is no evidence that the internal dynamics of the turbulence is aging, and I would be surprised if this were the case. I would like the authors to consider whether aging is really an appropriate description to use in the title of the paper. If so, then I would like to see some clearer justification for this term.

It is correct, that the small scales of stripes, i.e. streaks and rolls are not aging, at least not initially. However the process is also not solely a contraction of the stripe but as we will show/argue below, the secondary flow and the streak production at the stripe tip are involved. While the detailed mechanism is likely more complex the term aging was chosen based on the lifetime (splitting) statistics and we hope this aspect is clear from our data: The point is that the probability for a stripe to decay/die increases with stripe age. Or in other words the older the stripe the lower its life expectancy. This is in contrast to memoryless stripes/(puffs) where just by looking at such a stripe or puff at an instant it is impossible to define an age. A puff that has been created a much longer time ago has identical average properties and equally its probability to decay in the next time interval is identical to that of a young puff. To clarify this point we have added the following sentence to the end of the introduction: 'Instead of being constant in time, the probability of a stripe to decay/die increases in time and hence stripes age.'

To summarize, as the referee correctly points out aging does not refer to small scale turbulence but to the overall structure, i.e. to the stripe itself. In the previous version our usage of the term aging was not always consistent and misleading in some instances. We have corrected the sentences in question (including the title) and it is now explained that the term aging refers to the fact that the probability of stripes to decay (die) increases with the stripe's age (in contrast to the constant decay probabilities encountered in other shear flows).

In this context, we have also included a decomposition of a stripe's velocity field into large and small scales. As can be seen from Supplementary Fig.5, the large scale circulation around the stripe is what appears to slow down first, while the small scale fluctuations initially remain at a more constant level. This observation is also consistent with the observed slowing down of the downstream tip (Fig.3b). The streak production at the tip is driven by an instability of the mean flow around the head of the stripe (Xiao & Song 2019). While this quantity is not identical to the large scale circulation, they are closely related. Since from the experiments it is clear that the aging process is connected to a slow-down of the streak production, it is hence consistent that the large scale circulation around the tip slows down prior to the eventual decay of the small scale flow.

2) From the supplemental material: "Each stripe is considered at a definite time after the perturbation, its area determined and then an average found over all the stripes at that Re at the same time, thus yielding an ensemble average of the turbulent fraction at that time." I have concerns about this method of aligning the different samples. It is not clear that a fixed time after initiation will align the stripes to equivalent states. Why not align them based on a fixed value of the turbulence fraction or stripe length and look at the time to decay from that time? One would need to investigate the dependence on the value used to align the samples etc. One can already see in the DNS data for energy in Figure 2(b) that there may be some alignment issues. I would expect that using turbulent fraction rather than energy would smooth those decay plots.

We need to distinguish two different aspects here: 1) The analysis that involves the area of stripes in order to determine the growth rate. In this case, we intentionally use a typical distribution of different stripe sizes that naturally arise from a quench. We here want to get the average growth rate of this distribution of stripes and hence alignment to some fixed size would not be useful.

2) The stripe lifetimes. Here indeed the initial stripe length may play an important role, the underlying hypothesis would then be that the decay probability is determined by the length or overall size of the turbulent stripe. To probe this we have followed the referee's suggestion and aligned stripes according to the turbulent fraction. As shown in below figure this size alignment does not lead to a data collapse and actually does not considerably reduce the spread. Hence the decay is not solely determined by the stripe's size. We assume that rather than just the size, the initial conditions (after the quench) matter, such as, for instance, the magnitude of the large scale flow.

Time evolution of the turbulent fraction for the fully localized stripes shown in Fig.2b in the paper, with the curves being shifted such that they have the same turbulent fraction at $t = 0$.

3) A final point is related to the previous points. Consider the following. If the contraction of stripes occurred at exactly the same rate for all realizations and all stripes are aligned suitably to start with, then P_{survival} would be a step function, 1 until some common decay time and then 0 after. Now, due to intrinsic fluctuations of turbulence, and due to differences in initial conditions (and possibly issues with the alignment noted in 2), the decay times will not all be the same. Hence the P_{survival} will now be a rounded step function. The question is, are we seeing something fundamentally different from this? Is there a signal in the lifetimes that is different from just broadening due to the factors just listed? Again, is this "aging" in that there is some significance/universality in the form of the survival functions, or is it instead a broadened deterministic decay process? I appreciate that this might be difficult to address and so I do not

necessarily expect the authors to fully address this in the revised manuscript, but I will ask them to consider the question as it is clearly important.

We have partly addressed this point above. While it is indeed difficult to answer this point and to rule out that the survival probability curve P_{survival} could be a noisy step function, we think this is less likely. As shown, despite length alignment, the distribution remains quite broad and as we argue, the slow down of the large scale flow occurs directly after the quench and is involved in the aging process.

Moreover some of the energy curves (for example, see the lighter of the turquoise curves in Fig. 2b) do show a brief recovery and hence a non-monotonic decay. We therefore expect that the overall decay process has a stochastic component to it as well, with a certain probability that the decay in turbulent fraction slows down and the tip speed stabilizes for some time. With respect to the form of the survival functions, the new data closer to critical ($Re=630$) indicates that the rounding of P_{survival} becomes more pronounced as the critical point is approached. The overall shape hence is composed of the initial phase where decays are absent and this also increases as Re_c is approached. After this initial phase, the decay probability increases with time. We now describe the shape of P_{survival} in more detail in the manuscript.

Clearly a full understanding of the mechanism and the exact sequence of events involved in the aging process requires further studies and cannot be resolved in this first report of the phenomenon.

Minor comments

"Note that in all the experiments, stripes remained at a minimum distance of 50h from the lateral boundaries". I understand but suggest instead: "Note that in all the experiments, stripes remained at least a distance of 50h from the lateral boundaries".

Thanks. Indeed this was not well phrased, and we have corrected this

Figure 2(a): the authors should comment on why $Re=620$ from DNS is like $Re=600$ from experiments, or at least alert the reader. I was confused.

We now discuss this point. Since the stripe decay is not memoryless, initial conditions are likely to matter here. While we did not investigate such a possible dependence, it is likely that quantities like the initial size of the stripe, the nature of the quench and the initial magnitude of the large scale flow etc. may matter. Finally we cannot rule out that there may be a small discrepancy between experiment and simulation (either due to numerical resolution or experimental uncertainties).

Figure 3(c): maybe some faint lines separating the panels.

We have changed the figure accordingly.

"Aging dominates stripe dynamics below as well as above the critical point and both aspects are closely connected to the deterministic growth at the stripes' downstream tip, an aspect that is crucially missing in many previous studies of channel flow." Apart from the issue of the term aging, this is misleading in that there are several recent studies on the growth of stripes, and it is widely recognized that this will affect the nature of transition.

First of all we hope that the definition of 'aging' is clear now, i.e. it is referring to the increasing (with time) probability of a stripe to die (similarly above critical to split). Secondly we have shortened the

statement in question to: "Aging dominates stripe dynamics below as well as above the critical point and both aspects are closely connected to the deterministic growth at the stripes' downstream tip."

Figures 2 and 3 in the supplementary material: captions refer to (a), (b), (c) but there are no labels in the figures.

We have added the labels.

Reviewer #2 :

We thank the referee for the comments on our manuscript. We have taken considerable time and effort to obtain the requested data closer to critical. This required various changes to the set up and in addition due to a lab move we had to rebuild the entire experiment. This unfortunately caused a substantial delay. As the referee will see below, the new data (new Figure 2a), fully confirms the lack of memorylessness in channel flow and the aging of individual stripes. Here our response to all the points raised:

This paper investigates the time evolution of turbulent stripes around the critical point to turbulence in channel flow, and reports that the evolution is far from memoryless, the Poisson process. For this reason, the authors conclude that the theory of directed percolation (DP), which requires the Poisson process, is invalid to channel flow while many previous studies have shown the adaptability of the theory.

There is a subtle difference: the point we make in the manuscript is that 'the mapping between turbulent patches [i.e. stripes] and DP sites is not fulfilled.' Hence, what we show is that the analogy between individual active sites and individual turbulent patches that works in Couette and pipe flow does not apply to channel flow. In other words, our argument and observation are on the level of the microscopic processes and rules, and that in these aspects channel flow is qualitatively different from the aforementioned other shear flows. The conclusion we draw from this (see previous version of the manuscript) is that this 'challenges the robustness of the transition model'. However and this is the point we would like to stress, taken on its own this does not necessarily violate one of the DP conjectures and we also did not claim this. This being said, it is known from coupled map models that far more subtle changes can alter the universality class and lead to a 'non-DP' transition.

The experiments and numerical simulations in this paper are very huge and valuable, and I imagine the authors carefully analyzed the big data. Especially, Figure 4b is a very meaningful diagram clearly showing the characteristics of a turbulent stripe.

Although the presentation of experimental data is very good, some of their interpretations are fatally lacking in evidence. If the comments below are not misguided, I cannot recommend this paper to Nature communications.

(1) "The authors investigated only growth along turbulent stripes. However, there are several kinds of growth/decay processes in channel flow."

It is important to note that we do not restrict our analysis to only growth (or decay) along stripes but rather we determine growth as such, regardless of the underlying mechanism. The growth rate shown

in Fig.1 is determined by computing the turbulent fraction, not just of an individual stripe, but the total area covered by turbulence (e.g. split off pieces etc. are also taken into account). We have altered the text to make this clear. However, if we do look at the actual mechanisms involved, we indeed find that there are only two growth modes observable close to the critical point. This is (1) single stripe growth (i.e. elongation) and (2) splitting. Other growth processes such as branching (i.e. a new stripe grows and protrudes from an existing stripe) are only observed for considerably higher Re (≥ 1000), a regime that is not part of our study.

“Furthermore, the splitting rate at a given length (P_{split} in Fig.4b) may be memoryless. Since a rich variety of stochastic models exhibits DP behavior, they must show more clearly why the deterministic growth spoils the requirements of DP theory.”

We agree that P_{split} for a stripe of a given length, i.e. a constant size may be constant in time and hence memoryless. Even if this were the case, actual stripes undergo a growth and shrinking cycle in which the length is continually changing. This by itself fundamentally alters the nature of the stochastic splitting process. Note that models of shear flow transition like the one by Manneville and Shimizu (2020), assume Poisson statistics, an assumption that as our study shows clearly does not hold for actual stripes. Future models crucially must take these peculiar features of actual channel stripes into account in order to make predictions on the possible universality class.

(2) Although their experimental equipment is very huge, it is not still enough to obtain equilibrium single stripes.

We are not entirely sure how the referee defines the term ‘equilibrium stripe’. As our study shows, localized stripes are never in equilibrium but they constantly change their size; this is one of the central findings of our experiments. We assume the referee means the average stripe size at a given Re .

The size of our experiment is more than sufficient to not only capture a stripe of average size, but as shown in Fig.4b it is even large enough to capture the full range of sizes stripes exhibit during their lifecycle. To be more specific at this Re ($=710$) stripes have an average or median size of $\sim 250h$. Half the stripes did not reach this size and split before, and half grow larger prior to splitting. Stripes have an angle close to 45 degrees and hence a length of $250h$ (see Fig 4a) corresponds to a spanwise extent of $\sim 177h$. Even very rare events like the top 1% only have a length of $\sim 330h$. With the corresponding spanwise extent of $\sim 233h$, they are still significantly smaller than the channel width of $490h$. Of course at larger Re , stripes may grow to sizes larger than our channel but we are interested in the vicinity of the critical point and $Re=710$ is well above critical.

The aspect ratios (specially in the streamwise direction), and the corresponding observation times are by far the longest that have been realized in any experiment to date and this is precisely why we have been able to identify that stripes are not in equilibrium but go through a growth-splitting cycle.

The growth rate in Fig.1 makes sense only for specific initial conditions. Since the turbulent intensity along a stripe is not uniform, it has a finite equilibrium length which depends on Re .

To determine growth rates in Fig.1, we used two different methods to create stripes and as discussed in the manuscript, and the results do not depend on the way the stripe was created. The more robust method at lower Re is to initiate a stripe at higher Re followed by a quench. This procedure resulted in initial stripe lengths (i.e. just after the adjustment time following the quench when the image recording was started) ranging from 120 to 240 h (the range has a mild dependence on Re). This is hence a fairly wide size distribution. Moreover, and as becomes clear from Fig.4, it is the natural range of stripe sizes in this Re regime. Intact stripes tend to have a minimum size of 100 h (smaller segments tend to disintegrate), whereas above 250h the probability to split becomes large. Thus, the growth rates in Fig1

are calculated from an ensemble of stripes that have the typical size distribution encountered after a quench in this Re regime. Note that we can also split initial conditions into different size ranges, (e.g. the larger and smaller half) - they show the same growth rates.

(3) The word “aging” seems reasonable to me.

As can be seen from Fig.2b, the stripes' energy decreases gradually over time, i.e. the structure ages. This is in contrast to the memoryless, “non-aging” cases where the energy level fluctuates around a constant level and then drops suddenly. Correspondingly the decay rate for a memoryless process is constant in time and this is not the case for localized stripes. The older the stripe, the higher the decay probability and hence the shorter the remaining life expectancy. This is what we refer to as ‘aging’. To clarify this point we have added the following sentence to the end of the introduction: ‘Instead of being constant in time, the probability of a stripe to decay/die increases in time and hence stripes age.’

Also we removed misleading sentences from the previous version that stated that ‘turbulence ages’ and we clarified that what ages is the overall stripe. In this context we have also added a decomposition into small and large scales (SI Figure 5) which shows that the process is connected to the large scale around the stripe which slows down. Either way the point is that stripes structurally change in time and in this process the decay probability increases.

Since $Re=620$ is too far from the critical Re , the decaying trajectories in Fig.2 are inevitable. Previous studies observed DP behaviors only within the small range of Re . It is necessary to investigate decay/growth in the vicinity of the critical point to show the difference from other flows. Although the motion at the downstream edge of a stripe is close to periodic, a large deviation may abruptly turn it to decay. I have seen the numerical result of a single strip which decays after long transient $O(10^5)$.

The referee might have overlooked the point that we selected comparable distances from the critical point for the different data sets (pipe compared to channel). In Fig.2. $Re=620$ is in actual fact closer to the critical point than the two pipe Reynolds numbers - 620 is 5% below critical whereas 1860 and 1900 are 9% and 7% below the critical point of pipe flow!

We have nevertheless followed the referee's wish and obtained lifetimes considerably closer to the critical point. To do this unfortunately took a very long time because in addition to the substantial modifications to the channel, we meanwhile had been forced to move our lab to a different building. We eventually managed to rebuild the experiment and we now approach the critical point to within 3% ($Re=630$). Despite the close vicinity to the critical point, lifetimes remain non-memoryless. Interestingly the time before the first decays set in keeps increasing, and the tail that follows is non-exponential.

Although the motion at the downstream edge of a stripe is close to periodic, a large deviation may abruptly turn it to decay. I have seen the numerical result of a single strip which decays after long transient $O(10^5)$.

In our experiments of many thousand isolated stripes above the critical point we have not observed such an event, where a stripe decays abruptly. Of course we cannot rule out that in very rare events this may happen. Another possibility is that in simulations it may be possible that self-interaction via the periodic boundary conditions plays an additional role that is absent in experiments. Of course one also has to keep in mind that in experiments it is not possible to follow individual stripes for as long times as in simulations.

Nevertheless, besides the possible occurrence of such rare events, we hope that it is clear from our data, including the new measurements closer to critical, that the overall behavior of a typical stripe in channel flow is very different from that of localized turbulent structures in other flows.

Reviewer #3:

We thank the referee for the thoughtful comments. We fully agree with the central role that large scale flows (LSF) play in the context of turbulent stripes. However since our study focused on the lifetime and proliferation statistics, and only to a lesser extent on the underlying mechanisms, we had not included this discussion. Following the referee's advice, we have carried out the requested decomposition and added a brief discussion of LSF, and it actually fits nicely with the slow down of the tip activity we observe.

Prior to responding to the detailed points, we would like to mention that one of the other referees requested data closer to the critical point. We have therefore carried out lifetime measurements at $Re=630$ (see new Fig. 2). In order to do this we had to reassemble and substantially modify the experiment which took us close to a year to accomplish, and we apologize for this delay in our response.

So, it is fundamental to verify if the existence of this more homogeneous circulation, the LSF, that could be at the origin of many description made by the authors on the evolution of the stripes. It is the case, for example, to understand what is discussed in the Figure 3 and the long persistence of the stripes before the abrupt decay. It is very important because these elements could lead to the main conclusion of the study that is to say the absence of memoryless process.

We have now included a brief discussion of this aspect and the possible implications that arise from it. However we would like to stress that an in-depth study of the mechanism underlying aging is not the purpose of the present manuscript and has to be left for future studies. This being said we appreciate the referee's point and have investigated the LSF during the stripe decay.

To do so, and as suggested, we have analyzed the numerical data. Indeed in experiments due to the small channel height velocity data is not accessible. Given the very large aspect ratio of the experiment and the large stripe sizes (typically $200h$) it is equally not feasible to track tracer particles around an entire stripe. We decomposed the computed velocity fields into small and large scales and in the figure below, we show the respective energies for a decaying stripe. As can be seen, the small scale flow (SSF) initially remain at a more or less constant level, while the LSF energy already drops. The data indicates that the aging appears to start from the LSF and that once the driving via the large scale motions becomes too weak the stripe starts to disintegrate. This finding agrees with the important role of the mean flow around the stripe's tip which has been shown to drive the streak creation at this location (see Xiao and Song (2020)).

This figure shows two examples of decaying stripes. In both cases the LSF decay essentially from $t=0$, the end of the Re quench. Whereas the SSF remains more or less at a constant level for the following 400 advective time units and only subsequently starts to gradually decay. Eventually (for $t > 1000$) the SSF decay overtakes that of the LSF.

While it is of course important to eventually understand the aging process in even more detail, this requires a much more in-depth investigation which is a separate study in its own right. Together with a discussion of the suggested literature of LSF, we now state above observation in the manuscript and add the LSF-SSF decomposition to the SI. We hope the referee agrees that the LSF does not distort the scenario but rather they fully support the aging of stripes and its connection to the slow-down of the tip activity. We added the following paragraph to the paper:

‘The velocity fields of stripes in the direct numerical simulations can be further analysed and in particular and as suggested in a variety of recent studies [57-59], decomposed into small scale and large scale flows where the latter constitute a circulatory flow around the stripe. The mean flow around the stripe’s downstream tip – a quantity closely connected to the large scale flow - has also been investigated by [56] and was found to be central to the streak creation. Our preliminary analysis (see Fig.S5) appears to confirm this view. The large scale flow appears to decrease in energy before the small scale turbulence is affected, indicating that aging and the slowdown of the streak creation is hence connected to a weakening of the large scale flow. However a detailed understanding of the origins and the causal relation between the different process involved in stripe aging requires an in depth investigation of the flow fields of individual stripes and ideally a forcing and or filtering of different flow components, which is beyond the scope of the present work and left for later studies.

The above observation also relates to the referee’s next remark:

Another key question about the validity of the results exposed in the manuscript arises from another limitation of the flow visualization technique, which is not capable to distinguish between decay times for the different small-scale velocity components that is to say streaks and rolls as it

has been put in evidence recently by Liu (2021). These results need to be taken into consideration as the LSF can survive longer times, perhaps by viscous mechanisms (see Rolland (2015)). The LSF may distort the scenario of the aging of stripes. Once again, the possibility to revisit the existent numerical data of the DNS can be put light around these open questions.

As discussed above and shown in above figure, the decay of LSF actually starts earlier and this is what potentially starts the aging process. On the other hand if we consider the later stage of decay and as the most energy is contained in the largest scales, the smaller scales disappear faster, i.e. as can be seen in above figure the energies of LSF and SSF intersect at later times. A further decomposition of SSF into rolls (E_w) and streaks (E_u) is shown in the figure below (streak energy in black, roll energy in red), but

Time evolution of the energy of the small scale component of the fluctuation velocity in the streamwise direction (associated with streaks) and in the spanwise direction (associated with rolls).

at least as far as we can tell, this decomposition does not add further insights into the aging process so we have not added this to the SI.

The paper is well written. From this point of view, only few paragraphs deserve to be modified to gain clarity. In particular the last one in page 8 which refers to the influence of geometrical changes or imperfections which is not comprehensible for this referee. Is it related to the influence of noise or external perturbation as already discussed in Liu (2021)? The mention of geometrical differences

Geometrical changes and imperfections referred to observations in quasi-Keplerian flows that a modification of the end walls completely changes the stability of flows and likewise in pulsatile flows where minute curvature and small protrusions in the pipe wall introduce a new instability. Either way, to address questions by another referee we changed the wording of this part.

Regarding the minor points raised by the referee:

Cut out the irrelevant reference...

Done

Begin the references of DP and turbulence...

The reference has been added and we also added a recent review on DP and shear flow transition which amongst other things explains the differences between Pomeau's suggestion and the DP analogy found, for example, in Couette flow.

It is said that "In the transitional regime, turbulence appears in the form of stripes", but it is necessary to briefly discuss the fact that in PPF, Carlson (1982) and Lemoult (2014) have observed spots and that Tsukahara (2014) has observed bands.

The references have been added. We would like to mention in this context that in large aspect-ratio experiments, spots can only be created as short lived transient structures as they either decay or develop into stripes (now discussed in the manuscript).

In the assertion "1.5 times the bulk velocity ub..."

Thanks, changed accordingly.

p.4 and p.1 in supplementary information: As the initial perturbations to generate strips are obtained moving the ferromagnetic obstacle, how is the influence of his inclination motion across the channel?

We have tested different inclinations and did not find a big effect. If we define 90 degrees as the direction perpendicular to the wall then any angle between 60 and 120 degrees appears equally efficient in triggering stripes. While there may be some differences, at least from our present data there are no large changes.

p. 5, Fig. 1 and p.3 and 4 in supplementary information:

How is exactly measured the growth rate? It is necessary to show a figure with the data leading to this estimation (time evolution of the turbulent fraction). Is this quantity the inverse of a typical life time? If it is the case, these are not very long (only 10 units for $Re = 620$) There are only 3 values of the growth rate before the extrapolated value of Re_c . What are the criteria for choosing 4 values (and not 3 or 5, even 6)? Has the critical value be obtained by extrapolation of negative growth rate or by change of sign?

The procedure was briefly explained in the last paragraph in the section on image processing in the SI. This has now been expanded and moved to a separate section. We hope it is now clear that it is not measured as an inverse of a lifetime, but rather as the Re where the growth rate changes from negative to positive. The first four points (with growth rates close to or below zero) can be fitted by a straight line and the Re at which it intersects the zero growth rate line is then an estimate of Re_c .

On the one hand, it requires a considerable length of time to collect statistics from ~ 1000 stripes, also taking experimental difficulties like the settling of visualization particles etc. into account. On the other hand, we would also like to stress that the measure of growth rates is not a direct measure of the critical point but an estimate. Just like the intersection of the splitting and lifetime curves in pipes, also this method does not take interactions between structures into account. Given this circumstance we believe that the change from positive to negative growth rates is captured sufficiently accurately. Adding further points would not shift this value significantly.

p. 5: It is mentioned in the manuscript that the value of $Re_c = 650$ agrees with references [41 - 44], which seems to give higher value. How to compare with the equivalent DNS study in the reference [39]?

In the paper, we do not attempt a quantitative comparison with the references [41-44] (New reference numbers [49-52]). Rather, the point we make in the paper is that all these studies, like ours, show evidence that stripes begin to grow far below 830 (the critical point proposed by Sano and Tamai (2016)). We have now re-worded the text to make this clearer.

To be more precise, references [41] and [43] do not carry out a statistical analysis with sufficient realizations. However, they do find band continuous extension beyond $Re = 660$, with an eventual decay occurring only because of self-interactions due to periodic boundary conditions. [41] explicitly state "Based on the above observations, it is expected that the oblique extension will continue for $Re > 660$ if the computational domain is large enough..." Reference [42] shows non-zero steady-state turbulent fractions down to $Re = 700$. A linear extrapolation of the turbulent fraction to zero gives a value of Re_c consistent with ours.

p. 5 and Fig. 2:

In the reference [39] it is observed that a turbulent band persists as a long-lived metastable state before decaying abruptly. In consequence, his Figure 9 presents the evolution of spectral quantities and velocity field norms in function of time plotted with a translated time beginning when decay and exponential decay are observed. What happens if this procedure is also applied on the results of this work?

In our case we do not have only 1 or 2 dominant long wavelength modes, for localized stripes the energy is distributed among more modes than is the case for the periodic domain simulations. Hence in the figure below we show the longest wavelength modes of the streamwise velocity fluctuation that have significant energy content as well as the energy norms associated with the u, v and w. In [39] (new reference number [41]), in their study of non-localized stripes the decay indeed remains memoryless as discussed and hence, in the initial phase, the spectral quantities as well as the velocity norms have a constant level. By contrast, the spectral quantities in our case decay right from the beginning. Moreover we do not find a clear collapse of the curves from different runs. This analysis does not appear to add much further understanding to the aging process and we hence have not added it to the paper.

Following the procedure in reference [41], all runs are truncated at the same final value of u^2 , the quantities of interest (spectral quantities and velocity norms) shifted so that all realizations end at the same time and normalized so that they have the same final value. The left panel shows the time evolution of the energy of two of the longest wavelength modes of u for different runs at $Re = 620$. The right panel shows the velocity norms for the same runs. Unlike ref [39], we do not obtain a collapse of the different runs using this procedure and moreover, the spectral quantities decay right from the beginning.

p. 13, Fig. 3a and 3b: The spanwise direction is Z and not Y as is noted in the figures.

Well spotted. To be consistent with the simulations we changed the figure and now have y as wall normal and z as the spanwise direction.

Response to referee1:

We thank the referee for looking at our manuscript again and the additional question raised is indeed a very relevant one. Our response is given below.

Referee remark: As far as I know, there is no equivalent study of the decay process of unconfined stripes in any other shear flow unconstrained in two dimensions, e.g. plane Couette flow or unconstrained circular Couette flow. Hence, it is not clear that the nature of stripe decay presented in the manuscript results from considering channel flow, or from considering unconfined stripes. I have no doubt that the downstream growth mechanism of stripes in channel flow leads to fundamental differences with Couette flows. However, in the decay regime, the differences between the flows may be less important.

This is obviously a big question, potentially at least, but I want to the author to consider it. At a minimum they need to address this possibility and to clarify that all past studies of memoryless decay are for flow confined in all but one direction, or if they know of a memoryless decay in a flow unconfined in two directions, they can state that. Another option is for the authors to consider data from circular Couette flow (which they should have access to) and to extract decay statistics for stripes in that system. That would be considerable work. This issue is sufficiently important that I will insist that the authors address it, but I will leave it for them to decide how.

Author's Response: This is indeed a very important point and we should have included this in our paper all along. Regarding planar Couette flow we are not aware of any such data either, however, there is for circular Couette. In addition to two experimental studies of intermediate aspect ratio, which report memoryless decays (Borrero et al., PRE 2010 and Alidai et al., Flow turbulence and combustion 2016), a lifetime study in a much larger aspect ratio set up has been published more recently by Avila & Hof (Entropy 2021). In this latter study the circular Couette set up has a spanwise size $526h$ and streamwise $622h$ (h being half gap). The spanwise aspect ratio in that case is almost identical to that in the present channel experiment. In that sense the circular Couette stripes are as unconfined as the channel stripes we study. Lifetimes of (circular)Couette stripes were found to be memoryless as shown in below figure (Fig. 5 in their paper).

However, lacking a perturbation mechanism to create single stripes, these authors quenched from spatio-temporal intermittency. Patterns then quickly broke down into localized stripes, but initially often contained more than one stripe.

To determine if this influences the result we selected one of their data sets, $Re=530$ (red symbols) and reanalyzed the data, measuring the lifetimes from the point in time when only a single stripe remained in the domain. As shown in the figure below also for this single stripe case we obtain a Poisson distribution. (Note that this analysis is based on 50 individual stripes and hence the deviation below 10^{-1} can be safely attributed to insufficient statistics).

One observation we made while reanalyzing their data is that Couette stripes even below the critical point do not necessarily remain single stripes, but frequently branch and split. In channel flow below the critical point no such events have been observed for the available observation times. In channels in all instances single stripes remained single stripes below critical and splittings could only be detected above critical, i.e. when individual stripes grew persistently. In other words in channels splittings strongly depend on the elongation of individual stripes (see our Fig. 4c) which is not the case for Couette stripes.

We hope that the comparison to the Avila & Hof data answers the referee's question, i.e. that localized stripes in circular Couette flows indeed are memoryless we have added this reference [43] and a brief discussion on p. 3 of the manuscript.

Response to referee 2:

We thank the referee for looking at our manuscript again. We are happy to provide an in depth response to all the points raised:

Referee remark: *The size of isolated stripes, like turbulence in a pipe flow, fluctuates around a Re-dependent average.*

Author response: As we show in below figure, this statement is not correct for localized stripes. Puffs indeed fluctuate around a well-defined mean and they do this above as well as below the critical point. However, stripe sizes do not. We explicitly show this in below figures (puff data is taken from DNS in a 200 R long domain, stripe sizes are measured in our experiment). The first figure shows an example of the length evolution below critical. The puff fluctuates around a well-defined average size for the first ~900 advective time units (black is the original puff data, grey is the same data scaled up by a constant factor), subsequently it decays. In contrast, the channel stripe (red curve) shrinks with time. At no point does it fluctuate around an average size. This is already shown for a different quantity, the energy time series in Fig. 2 in the main manuscript.

Above critical as shown in the second figure the puff (black curve) again has a well-defined average size whereas the stripe (identical data to figure 4b) grows at a constant rate then a macroscopic piece splits off, it grows, splits etc. This is precisely the cyclic behavior we describe in the paper and illustrate in the current Fig. 4c (previously 4b). These are not puff like small fluctuations around some average. Instead underlying the cyclic dynamics is the constant growthrate at the stripe tip. This mechanism sets the slope of the ‘upward ramps’ visible in the figure. No equivalent mechanism exists for puffs. Earlier DNS (discussed in the introduction) reported expanding stripes at these Re (as well as the tip instability) but domains were far too small to observe the splittings and the overall cyclic dynamics.

We have added above panel (stripe length vs. time above critical) to figure 4 which we hope more directly shows the stripe growth-splitting cycles. We trust that this clarifies the fundamental difference to puffs in pipe flow.

Referee remark: *This is already a well-known fact. It has been reported in several papers that a stripe survives at least about $10^4 h/U$ at $Re=700$ while keeping its average length.*

Author response: Unfortunately the referee does not provide references here. There are indeed a couple of numerical studies by Tao and co-workers that simulate single stripes for 10^4 advective time units (h/U_{cl}), Xiong, Tao, Chen, Brandt PoF2015 and Tao, Eckhardt, Xiong PRF 2018. As the referee states these authors report that stripes survive for 10^4 time units at $Re=700$ keeping their average length as shown (for energy) in the left panel of below figure (green curve in panel (d)). However, it is apparent from the right panel (2) that these are not localized stripes. As the snapshot at $t=5200$ shows this stripe connects to itself via the periodic boundary conditions and from this point it simply cannot grow any further.

After the stripe ends connect there is no tip instability anymore and such periodic stripes fundamentally differ from the naturally localized counterparts. As we show in the paper such artificially periodic stripes do not age (see blue data points in figure 2a of our manuscript). All of this is caused by the insufficient domain size and periodic boundary conditions.

To our knowledge, our experiment is the first study to date that has a sufficient aspect ratio (4000 x 490 in contrast to 200 x 196 by Tao et al.) to study the dynamics of isolated stripes.

We have added a paragraph on p. 3 that discusses the issue of stripe self-interaction in periodic boxes.

Referee remark: *In this sense, the authors' experimental domain size, which corresponds to an advection time of 1000 h/U, is still too small to consider their main claim.*

Author response: This is incorrect, please check sizes and time scales given in the paper: Our domain size is streamwise 4000h. This length corresponds to 6000 advective time units. If we use the same time scale to non-dimensionalize our data as customary in simulations (e.g. by Tao and co-workers), h/U_{cl} (Where $U_{cl} = 1.5 U_{bulk}$) and because stripes advect at a speed close to U_{bulk} , *the experimental domain size corresponds to: 4000 times 1.5 and hence 6 000 advective time units (h/U_{cl}).* See our new figure panel 4b. **Our observation time is hence 6 times larger than what the referee claims.** Even though this may appear slightly smaller than the 10 000 h/U_c in the Tao simulations, as discussed above, their physical domain size, i.e. the aspect ratio, is the limiting factor and stripes do not remain localized.

Referee remark: *Figures 1 and 2 may only represent the initial transient period due to the perturbations.*

Author response: Since the referee underestimated the time scales in our paper by a factor of six, we assume this remark obsolete.

Moreover we extensively discuss the role of initial transients, see SI. Out of the 6000 advective time units available, we typically leave the first 2250 advective units (recordings usually start 1500 h from the inlet) out precisely to allow stripes to fully develop and initial transients to decay. We believe this to be a rather careful and conservative choice that assures that what we observe are not transients. Finally the remaining observation time of over 3000 advective units is still considerably longer than any experimental study and comparable if not longer than the times over which stripes remain truly localized in aforementioned simulations.

Referee remark: As to the distance from the critical point, the comparison with the pipe flow is not meaningful.

Author response: The comparison we carry out is factual and straightforward. We state that at the same relative distance from the critical point, in one geometry decays are memoryless, while in the other they are not. The referee does not specify what she / he believes to be wrong with this? Unless the referee thinks that the critical point determined for channel flow is wrong? We will answer to this below.

Referee remark: *To clarify their main claim whether stripes are memoryless or not they have to investigate stripes for $650 \leq Re \leq 700$ using much larger domain.*

Author response: As we point out in the manuscript and as well as in our previous response, **there are no decays for $Re > 650$.** The referee is asking for something impossible, how can we conduct lifetime studies if there are no decays? In fact it is well documented in the literature that above $Re > 650$ stripes continue to grow and do not decay. This has been pointed out by Xiong et al. PoF 2015 and Tao et al. PRF 2018. We here show Fig. 4 by Xiong:

FIG. 4. (a)-(c) Time series of E_k at different Reynolds numbers. Different colours indicate the simulations with different initial flow fields, which are the solutions of $Re = 665$ (black line) at $t = 0, 800, 1000, 1200, 1400, 1600,$ and 1800 . The horizontal grey bars denote the critical E_k for the ITB to survive.

At 650 (at latest at 660) any stripe of sufficient initial size/energy (in the figure marked by the grey bars) continues to grow. So far this fully agrees with our observation, however Xiong did not find any stripe splittings until $Re > 1000$. This is the central difference to our work and this led them to an entirely different interpretation of their data. They proposed that for $Re > 660$ a regime of ‘sparse’ turbulence is encountered where stripes survive indefinitely and continue to extend. The critical point they defined, is based on their observed onset of splittings at around $Re=1000$ where turbulence then spreads in both planar dimensions. While this definition is problematic from a statistical physics perspective, let's focus on the question why they did not find splittings for $Re < 1000$ whereas we do.

Well, also here the answer is the domain size. In their case the domain width of 160 h is the limiting factor. If we assume stripes to have a ~ 45 degree angle the maximum length that fits into this box before stripes reconnect via the periodic boundary is 160 h divided by $\cos(45) = 226$ h. In reality already somewhat shorter stripes will start to self-interact. Now let's look at our study: As we show the splitting probability depends on the stripe size. Let's specifically look at Fig. 4a, shown below. We have added a red arrow highlighting the stripe length at which the first notable splittings occur (for $Re=710$ in this case). As shown stripe splittings are only detectable for stripe length > 224 h. Hence Xiong et al. could not possibly observe stripe splittings in this Re regime. Their domain size does not permit the required stripe length. This equally explains why we are the first to observe the cyclic behavior of growth and splitting illustrated in Figures 4 b and c.

We have added a paragraph to the conclusion section that explains these key limitations of earlier studies and why in their domains they could not possibly find the full dynamics.

In summary, also regarding our observation that turbulence becomes sustained above 650 there is no conflict with previous studies. On the contrary our experiments accurately agree with these computer simulations in that stripe expansion persists from this point on. There are no decays above 650 and instead splittings occur for sufficiently large stripes, larger than permitted in earlier studies.

It is somewhat puzzling that the referee questions our results on the basis of the domain size, given that our domain size is the largest that has been used to date. It is precisely the fact that our domain size substantially exceeds that of earlier studies that allows us to for the first time capture the full dynamics of channel stripes.

Response to referee 1:

First of all we would like to thank the referee for initially recommending our manuscript for publication. The editor informed us of a concern that arose from the report by another referee, referee2. While the editor did not give us any further information of the nature of this concern we assume it is connected to the second referee's statement: *The biggest point of controversy between the authors and I is whether a turbulent band in channel flow is "aging" or "fluctuating"*.

There are two aspects to this statement. The first is about terminology and the question if "aging" is the correct term to describe the lack of memorylessness of channel stripes. This is a question referee1 raised in her/his first review and we will get back to this point below.

The second aspect is that referee2 uses 'fluctuating' and 'aging' as placeholders for two fundamentally different types of dynamics and she/he claims that what we observe in our experiment are transient dynamics: *"No matter how large the experimental setup is, it is difficult to achieve the "fluctuating" state in the present experiment."* Obviously if this was correct the main message of our manuscript would be wrong so allow us to clarify this second aspect first.

More precisely referee 2 is of the opinion that after some time the deterministic growth of stripes will stop and they enter this supposed 'fluctuating state', where the stripe length, like for puffs in pipe flow, fluctuates around an average size without splitting for excessive times (she/he proposes $>10^5$ advective units).

Let us also mention that this opinion of referee2 is not only in complete opposition to our experimental findings but in particular also to the recent work on the inflectional instability by Xiao and Song 2019, which identified the mechanism for the persistent growth mechanism at the stripe's downstream tip. Either way, in the last review referee2 has eventually provided the references that according to her/him show this 'fluctuating state' (figure 5.1 of Kanazawa's PhD thesis and the movie for $Re=725$ by Shimizu & Manneville PRF 2019) and as you will see below based on these studies this point can be easily clarified. Instead of disproving our study, the two simulations unambiguously confirm our experiments.

Before we show the data, allow us to summarize the second referee's opinion by quoting directly from the reports:

"...but I still have doubts about the main claim of "deterministic growth of stripes"

"The size of isolated stripes, like turbulence in a pipe flow, fluctuates around a Re -dependent average. This is already a well-known fact."

"A single band at $Re=700$ in a very large domain can survive longer than 10^5 (H/U) without splitting and decaying."

For simplicity we summarize the two positions in the left panel of below figure: the 'fluctuating pipe-turbulence-like state' is shown in grey (data from a puff, upscaled for comparison). The cyclic stripe dynamics observed in experiments are shown in red. The characteristic 'saw-tooth shape' of these cycles results from the continuous growth at the downstream tip and sudden splitting events. We determined the stripe length displayed in the Shimizu&Manneville movie for $Re=725$ and the data is shown in the right panel below (for a representative time interval, the inset highlights one splitting case):

It is apparent that the second referee's proposition regarding the dynamics of the simulated stripes is incorrect. The data evidently shows that stripes exhibit growth splitting cycles (following a saw-tooth shaped curve) just as they do in our experiment.

We further verify the excellent agreement between these simulations and our experiments by plotting the Shimizu Manneville data on top of our own data in the new version of figure 4c (see the latest version of our manuscript).

The second reference that referee2 provided, which again supposedly should show puff-like fluctuations of stripes, is Kanazawa's PhD thesis. We have hence kindly asked Genta Kawahara, Kanazawa's supervisor, for the respective data close to $Re=700$, but unfortunately Prof. Kawahara could not find the data anymore. He did however provide the following remark: "The longer stripe tends to leave its upstream segment, leading to immediate reduction of its length. This is consistent with the red dots in our data as well as your observed sharp decrease.

I remembered that this behaviour has also been observed around $Re=700$ (strictly $Re_m=470$)."

Remark: 'the red dots' refer to the panel 5.1(a) in Kanazawa's thesis for $Re=660$ ($Re_m=440$), which we reproduce in the figure on the right. Also this dataset shows the deterministic growth (upward slopes) as well as the sharp decreases due to tail splittings, or as Kawahara put it 'leaving of the upstream segment'. Hence from Prof. Kawahara's statement the growth splitting cyclic dynamics were equally observed in their simulations at $Re=700$.

The difference between 'fluctuating' puff-like and the cyclic channel stripe dynamics becomes equally apparent if we consider the distributions of size changes, shown in the newly added SI Figure 6. This quantity clearly separates fluctuating dynamics of puffs, which as expected show close to symmetric distributions centered around zero, from distributions of isolated channel stripes, which are strongly asymmetric. This asymmetry reflects the difference between the persistent tip growth and the abrupt shrinkage due to splittings (see also the added brief discussion on page 10 of the manuscript).

We hope that this lengthy explanation clarifies that the data of all studies of isolated channel stripes (Shimizu-Manneville, Kanazawa, our experiments and also Xiao & Song's JFM2019) perfectly agree on the persistence of the deterministic tip growth and the resulting splitting cycles. This settles the "point of controversy" raised by referee2 and shows that the assumption of a puff-like fluctuating (memoryless) state that referee2 had in mind is simply not applicable to channel flow.

Now back to the term 'aging' and this point is directly related to a question referee1 asked in the first reviewing round: *'The latter is clearly not memoryless, but in what sense is the turbulence aging?'*

The lack of memorylessness is shown in our Fig.2 . Consequently the life expectancy of an individual stripe is not constant but decreases with time. This is the literal meaning of the term "aging", we refer to the opening line of the abstract of a recent statistical mechanics paper on this topic: "Aging means that as things grow old their remaining expected lifetimes lessen."

(<https://www.sciencedirect.com/science/article/abs/pii/S0378437117305964>)

On the other hand we appreciate that 'aging' appears to also carry other implications that are not intended by us. To prevent any distraction from the main message of our manuscript we have removed the term 'aging'. This change allows us to keep the focus on the lack of memorylessness of channel stripes and its implications for transition, without risking unwanted interpretations due to terminology. We hope that our explanations and changes satisfy all the points and concerns of the referee and we again greatly appreciate the time and effort taken by the referee to evaluate our study.

Reviewer #2 (Remarks to the Author):

Response: We thank the reviewer for eventually providing the references to the simulations that extend for time scales of order 10^5 and that supposedly disagree with our experiments. Fortunately these references contain sufficient information to perform a direct comparison to our experiments and to establish if these simulations either prove our study wrong or confirm it. The referee's criticism was as follows:

*The biggest point of controversy between the authors and I is whether a turbulent band in channel flow is "aging" or "fluctuating". See the Figure 5.1 in Kanazawa's PhD thesis (...), where bands at around $Re=700$ ($440 < Rm < 480$) persist for a long time, and they can define an average length at each Re . Also see (...) *PhysRevFluids*.4.113903) which shows that band persist at least for a few 10^4 (H/U).*

Allow us to illustrate this "point of controversy" in below figure, left panel. **Possibility 1:** stripe lengths "fluctuate" around a well-defined average. Or as the referee had put it in her/his earlier review: "*The size of isolated stripes, like turbulence in a pipe flow, fluctuates around a Re -dependent average. This is already a well-known fact.*" Hence the 'fluctuating state' of stripes should qualitatively resemble that of puffs in pipe flow, which corresponds to the **grey curve** (puff size is rescaled by a constant factor for comparison). **Possibility 2:** in contrast to above, we find in experiments that the stripe length continuously increases until it sharply drops when the stripe tail splits off. These cyclic dynamics result in a characteristic saw-tooth-shaped time series, **the red curve**. Remark: "Aging" refers to the observation that during this cyclic process the splitting probability is not constant (it increases with the stripe's length).

The referee then reiterates her/his view by saying that these simulations show that "*A single band at $Re=700$ in a very large domain can survive longer than 10^5 (H/U) without splitting and decaying.*"

We will first look at the extensive study of Shimizu & Manneville (...) *PhysRevFluids*.4.113903) to establish if stripes indeed fluctuate and splittings are extremely rare, or conversely, if the computed stripes follow the cyclic dynamics we find, where the deterministic growth and sudden tail splittings give rise to a saw-tooth shape. From this movie it is straightforward to determine the stripe length as a function of time and we display a representative time interval in the right panel below:

Evidently the computational stripes show continuous growth and tail splittings, which results in the characteristic saw tooth shape. Clearly they do not assume a puff-like 'fluctuating' state. These cyclic dynamics persist throughout the entire simulated 10^5 advective time units. Note that due to the limited domain size not more than two stripes can co-exist in these simulations. The periodic boundary conditions introduce unwanted (self-)interactions that cause the split off tail to prematurely decay (or kill and replace the neighbouring stripe). Regardless of this computational limitation, the important point for the question at hand is that tail splittings are abundant (approximately one every 1000 advective time units). In light of this numerical (let alone our experimental) evidence the referee may agree that unlike what she/he proposed earlier, stripes do not keep "their average length" for extensive times, neither for 10^4 (first report) nor 10^5 (3rd report) advective units. The stripe length does not fluctuate. Stripes persistently grow until the length abruptly drops when a tail splitting occurs. These dynamics are strikingly (see left panel above) different from the 'fluctuating-state' in pipe flow. The key point of course is that, as we demonstrate (Fig.4a), the probability for the tail to split is not constant in time but increases with stripe length and this results in the lack of memorylessness.

Because the referee also questioned the deterministic/constant growth of stripes in one of her/his earlier reports (*I still have doubts about the main claim of "deterministic growth of stripes"...*), allow us to also remark on this point. For our experiments we visualized this deterministic growth in Fig. 4c by resetting time after each splitting, the resulting diagonal then corresponds to the growth rate. For comparison we have plotted the computational stripes of Shimizu&Maneville in the same manner and added them in the new figure 4c (red lines). The computational stripes, without any exception, follow the exact same slope, hence the doubt the referee raised, is settled by this comparison, the data speaks for itself.

We intended to also include the Kanazawa data (first reference provided by the referee) to this comparison and in order to do so we asked the supervisors, Professor Genta Kawahara, to kindly share the data for $Re \approx 700$ with us. Unfortunately Prof. Kawahara did not find the data anymore. He did however provide the following comment regarding this data:

"The longer stripe tends to leave its upstream segment, leading to immediate reduction of its length. This is consistent with the red dots in our data as well as your observed sharp decrease. I remembered that this behaviour has also been observed around $Re=700$ (strictly $Re_m=470$)."

Remark: 'the red dots' refer to the panel 5.1(a) in Kanazawa's thesis for $Re=660$ ($Re_m=440$), which we reproduce in the figure on the right. Again the data shows the characteristic saw-tooth shape and these

dynamics are equally confirmed by Prof. Kawahara when he refers to *"immediate reduction of its length"*. Note that Prof. Kawahara also implies the length dependence of the tail splitting process when he says *"the longer stripe tends to..."*. Now regarding the average lengths plotted by Kanazawa in his figure 5.1 panel b which the referee saw as evidence for a fluctuating state. It is clear from the time series in panel 5.1a, that these averages were taken over the entire time series which contains many tail splittings and growth-splitting cycles. Hence bands do not *"survive (...)* *without splitting"*, the structure of the stripe does not remain intact during this sequence but the tail repeatedly splits off. Even between splittings the stripe length does not *"fluctuate"* around an average, instead the length grows monotonically (*"deterministic growth"*), just as it does in experiments, and again the dependence of the probability on length is what makes the difference.

For completeness we also determined the size variations for above data, more precisely we are plotting the distribution of the size changes that occur on intermediate time scales (at time steps of 100 advective units) for the respective cases. The 'fluctuating state' is shown for comparison on the top left and has been obtained from a simulation of a puff in pipe flow at Re 2100. This latter distribution is centered around zero and close to symmetric as expected for turbulent puffs. This is in stark contrast to the distributions of the three studies of channel stripes, (Shimizu&Manneville, Kanazawa and our study). Here the distributions are strongly asymmetric. The sharp positive peak results from the deterministic tip growth, whereas the distribution is flat on the negative side and the entries here correspond to tail splittings which can reach very large values. We have included this figure in the SI and we added a discussion clarifying this distinction between the two types of behavior on page 10 of our manuscript.

In summary the two simulations unambiguously confirm the growth-splitting cycles we propose. This settles the point of controversy raised by the referee. The characteristic dynamics of channel stripes, driven by the continuous growth, is precisely what gives rise to the time dependence of stripe statistics (lifetimes, splitting probabilities) as we demonstrate in our study. This in turn violates the memoryless property, which prior to our study had been taken for granted as the key property of localized turbulent structures. At this point allow us to remark on terminology. Regarding the statement that "*The biggest point of controversy*" is if stripes are "*aging*" or "*fluctuating*". We would not have phrased it quite this way. The biggest point is if stripes are memoryless or not. While a decrease in life expectancy with time (our Fig. 2) is the literal meaning of aging, we appreciate that this term may carry other implications that are not intended by us. To avoid misunderstandings we have decided to remove the term "aging" from the manuscript and we directly refer to the time dependence of stripe properties and we hence keep the focus on our core message: the lack of memorylessness and the implications this fundamental change has for the transition to turbulence.

Reviewer #2 (Remarks to the Author):

First of all, we would like to thank the referee for taking the time to review our manuscript again.

Reviewer remarks:

1) As for the biggest claim, "memory of bands," I still disagree. This point regarding terminology (memoryless versus aging) has been made in the previous reports, e.g.: "The word "aging" seems reasonless to me".

There is actually a straightforward answer to this issue. The framework of population ecology provides exact definitions on the terminology of memorylessness and aging and a quantitative way to discriminate between these fundamentally different types of lifetime distribution. Applying this analysis to our data now provides an objective answer to the question of whether or not the channel stripes are aging. The standard method to categorize lifetime distribution is to fit a Weibull distribution (equation 1 in the manuscript) to the lifetime curve. Aging or non-aging is decided by the exponent k , known as the shape factor. As explained in the manuscript (see also fig. 2a) population ecology distinguishes three types of survival curves, type1: $k > 1$ (positive) aging, type2: $k = 1$ non-aging (=memoryless) and type3: $k < 1$ negative aging.¹

As shown on the right (new version of the Fig. 2b) this framework precisely matches all the data, the lifetimes of our channel stripes in experiments and simulations, those of puffs in pipe flow (notably with much shorter lifetimes) and those of artificial periodic channel stripes in computations (for the latter see Fig. 2c). As is apparent from the shape of the curves and quantified by the corresponding shape factors given in table 1, those different data are cleanly separated into type 1 (positive aging) and type 2 (non-aging, i.e. memoryless). According to this analysis, localized channel stripes are very evidently aging with k of around 8, whereas even much shorter lived puffs are clearly memoryless with a shape factor close to one.

We hope this analysis clarifies that for the presented lifetime data our use of the term aging is not "reasonless" but it is in accordance with the standard classification of lifetime analysis. Moreover in other flows (pipes, periodic channel simulations) even at the lowest Re at which lifetimes can be reasonably measured the distributions to all intense and purposes remain memoryless.

2) First, regarding the survival probability in Fig.2, it is obvious that bands decay monotonically at around $Re=600\sim 630$. This is because the downstream end cannot be maintained for this range of Re.

The referee actually raises multiple points here that we will address one by one. First of all the referee still appears to miss the point that decays of stripes can only be detected in this Re range, there are no decays from Re 640 onwards. Note that this does not happen for any other shear flow. Please have a

¹ For definitions and example see e.g. <https://www.nature.com/scitable/knowledge/library/survivorship-curves-16349555/> and https://link.springer.com/chapter/10.1007/978-3-031-28238-6_5

look at Avila et al. 2011 for puffs far longer lifetimes can be measured beyond the critical point. In this context the referee should also note that Avila et al. do not have a much longer observation time than we have (3500 advective units in their case). This we will cover in detail in point 3) below where we discuss the required time scales, but let us just point out here, that the majority of lifetime studies in pipe flow have far shorter observation times, often less than 1000 advective time units. The key point is that all these studies rely on censored data.

Regarding the it is obvious that bands decay monotonically, the referee still appears to be of the opinion that, despite our long observation times of up to 6000 advective time units, the entire process is some inevitable viscous decay process. Or as he / she puts it in an earlier review: *Since $Re=620$ is too far from the critical Re , the decaying trajectories in Fig.2 are inevitable.*

The referee may want to reconsider this, given that the lifetimes follow a Weibull distribution. This per definition is a stochastic process. Not a memoryless one, but one that falls into the category of ‘positive aging’. Allow us to show this even more explicitly, stripe lifetimes can be decomposed into an initial stochastic part and the eventual ‘inevitable’ viscous decay:

Although the decay of the total stripe energy is indeed monotonic (Fig.2d top panel), as we point out in the manuscript this quantity is dominated by monotonic decay of the large scale flow around the stripe. Conversely the local turbulence activity, i.e. the energy of the vortices and streaks that make up turbulence, is represented by the small scales of the energy decomposition which we had previously introduced in the SI (this follows from the studies of Westfreid, Duguet, Schlatter and others). As we now explicitly show in the lower panel of figure 2d, the local turbulent activity, so the energy of the small scales normalized by the stripe area fluctuates around a constant level (colour trajectories in below figure), prior to the eventual viscous decay.

We hope this analysis makes it evident that the turbulent structures within the stripe do not follow some monotonic viscous decay. They are sustained /alive for a well-defined time and then die. The distinction between life and death in fact is as clear-cut as in the case of periodic stripes (simulated for $Re=750$ for the box of Gomé, Tuckerman, Barkley PRF 2020), which are shown in light grey in above figure (lower panel of Fig. 2d in the manuscript). In both cases the eventual monotonic decay of the streaks and vortices composing turbulence takes about 200 advective time units, regardless if stripes are fully localized or periodic.

We would also like to emphasize that the periodic stripes shown in the lower panel of Fig. 2d have a shorter characteristic lifetime than the fully localized ones. Despite this shorter average lifetime the decay, as expected for a memoryless process, can occur at any time with equal probability. The only ‘inevitable’ part is the stripe death which corresponds to the eventual viscous decay and this is well distinguishable from the stochastically distributed lifetimes.

Of course the central difference between the lifetime distribution of the colourful curves and the light grey ones is the one between a type 1 and type 2 Weibull process. For the localized stripes all decays happen **at late times and within a relatively short interval**. This is precisely the expectation for a Weibull distribution with a large k value, i.e. a (positive) aging process in accordance with the standard theory.

Before we get to the referee's request of observation times of 10^4 to 10^5 , let us compare the actual lifetimes plotted in Fig. 2 for the different flows:

Again it is equation 1 that now allows a quantitative comparison of the memoryless puff decays and of the positive aging (type 1) stripes. The analysis of all the data is summarized in table 1, and as the referee can see the characteristic lifetimes, τ , of our localized channel stripes actually go up to 2300 advective time units, whereas the ones we are comparing to in pipe flow are as short as 40 advective time units, periodic channel stripes (Gomé et al.) are around 400. Following the referee's logic for these extremely short lived cases that we compare to, decaying trajectories should be all the more "inevitable". However for these other flows the shape factor (see table 1) remains close to 1.

We therefore conclude from this analysis (p.3 of the manuscript) that: "Even for characteristic time scales exceeding those of clearly memoryless cases in other flows by an order of magnitude, [...] stripe decay instead is a type 1 aging process."

We hope that the referee agrees that this is a factual statement.

3) Much longer time scale $O(10^4-10^5)$ experiments around $Re=700$ are needed to determine memory or memoryless. In this sense, it is impossible to answer this question with the size of the authors' experimental apparatus.

It is of course correct that in pipe flow characteristic lifetimes of 10^7 have been resolved which is far longer than the characteristic times we can measure, but the reason for this difference is not that our observation time is shorter than those in pipes. In actual fact, our observation of single stripes is longer than the observation time of single puffs in most lifetime studies in pipe flow.

The observation time of 10^4-10^5 the referee is requesting has never been realized in any pressure driven experiment. In pipe flow, amongst the longest observation times were obtained by Avila et al. Science 2011 and Hof, et al. PRL 2008, which achieved ~ 3500 advective time units (D/U). But they also only measure at a single location if the puffs survived or decayed, and they have no information on when the puffs decay. A lifetime study in pipe flow that is more similar to ours is that by Kuik, Westerweel and co-workers (JFM 2009) and we show their result on the right. The important point to note is that their observation window (see x-axis) is between 100 and 400 pipe diameters and hence it is only 300 advective time units. Despite this very short window Kuik et al. determined characteristic lifetimes of 10^6 and this is one of the most precise studies of pipe lifetimes, perfectly confirming other numerical and experimental studies. So how can Kuik et al. with an observation time of 300 advective units determine characteristic lifetimes of 10^6 , whereas we with our

much longer set up and an observation time of up to 6000 advective units cannot do the same thing in channel flow?

In pipe flow all studies use censored data. This means that nobody observes puffs across their full lifetime from birth to death, but single puffs are typically only observed for a comparatively short time (e.g. 300 units by Kuik et al). For memoryless processes, data censoring works perfectly and excessive half lifetimes (e.g. nobody has observed a single uranium atom in excess of a billion years) can be obtained with comparatively short observation windows just by looking at a very large number of cases (puffs, atoms etc.) .

We of course used the exact same approach and tried to determine much longer characteristic lifetimes at higher Re. Indeed we did many thousand measurements at Reynolds numbers all the way up to $Re=700$. If lifetime distributions were memoryless we should see decays even beyond the critical point (regardless if the exact critical point is 650, 660 or even slightly above), but in channel flow there is simply nothing to measure, there are zero decays for $Re>640$.

So what is amiss here? Why does the same technique that perfectly works for Kuik et al., for Avila et al. and all other studies in pipe flow, where for modest observation times of single puffs excessive characteristic lifetimes are measured, fail in channel flow? Again this is answered by the framework of population ecology and the different categories of survivor functions:

Let's pick $Re=660$ in channel flow, here we measured 500 stripes, and in this case we had limited our observation window to the last 2000 advective time units of the channel (which is still considerably longer than Kuik et al.'s observation window). Now since we know the characteristic lifetimes of channel stripes for Re 600, 620 and 630 (see table 1), we can estimate the expected value for 660 based on an exponential fit to those lifetime values. According to this τ should be of order 10^4 . The exact value does not matter, in either case this example will illustrate the fundamental difference.

If at this higher Re lifetimes were memoryless ($k=1$), which is what the referee suggests, then equation 1, for $k=1$, predicts the number of expected decays for the measurements we conducted. Let us quantify this: for $\tau=10^4$, $Re=660$ and an observation time $t=2000$ equation one gives $P=0.82$ and hence for the 500 runs conducted we should have observed around ~ 90 decaying stripes.

Now let us do the exact same computation with equation 1 but this time instead of memoryless and hence $k=1$, we use the shape factor k measured at lower Re . In other words, just as for the lower Re , we assume the survival function is of type 1 and $k = 8$. With all other numbers being exactly the same as above, equation 1 now predicts a decay probability of $P=0.999997$. Hence out of our 500 runs the probability to find a decay is factually zero and this is exactly what we measured, no stripes decay. The central point is that young stripes do not decay for aging processes and as τ increases with Re and exceeds our single stripe observation time.

We hence do not only base our observation of aging on the apparent aging survival curves at $Re=600$ to 630 where we can measure them explicitly, but equally on the fact that there is nothing to measure at higher Re . If stripes returned to memoryless at any higher Re , they would decay with constant probability, independent of their age, and looking at sufficient numbers of stripes we would see decays, but we do not.

4) Regarding the remark: The authors also associate memory with the non-exponential tail in Fig.4a, but this is meaningless. Fig. 4b is similar to the length time series in Kanazawa's paper that I sent in my previous comment, but it does not necessarily show "successful splitting" when a band becomes shorter.

For example, a single band is maintained for a long time period $O(10^4)$ at $Re \sim 700$ during repeated stretching and shrinking. A successful split is when a new band is formed stably next to the old one.

This is also the case for pipe flow ($Re \sim 2200$). A puff grows in length and splits by chipping, but the downstream one does not necessarily survive as a new puff. "The success or failure of a split is memoryless." A puff grows in length and splits by chipping, but the downstream one does not necessarily survive as a new puff. If it disappears immediately, we do not call it splitting.

First of all there is an issue with terminology here; we defined 'split' to mean for the stripe to literally split into two macroscopic pieces, irrespective of what happens to these pieces subsequently. This process is rather different from puff splits. See Figure 2C in Avila et al. Science 2011. Their space time plot shows that the puff does not split into two macroscopic pieces. In this figure the red regions of the two puffs do not connect, instead the new puff grows at a distance from the original puff from a small seed. (Avila's Fig. 2c shows this, the new and the original puff are separated by a blue, i.e. low energy, region). This is also precisely what Avila et al say, we quote: "puff splitting", in which new puffs are seeded downstream of existing ones...". We would like to point out that the referee's statement 'A puff grows in length and splits by chipping', is not how Avila et al. see this process, they imply that a seed escapes from the parent puff and grows after separating from this parent (again see their Fig. 2c). The referee may also consider that the size distribution of puffs is symmetric as we demonstrated previously in our Fig. SI 6. Growth splitting cycles however cause the size distribution to be highly asymmetric as, shown for the localized stripe case again in our Fig. SI 6. It is evident that the two distributions in Fig S 16 (panel a-c on the one hand and panel d on the other) are qualitatively very different we assume the referee agrees on this? Well, this difference reflects the difference of the two splitting mechanism.

Either way we are not comparing the splitting mechanisms for puffs and channel stripes in our manuscript, so above paragraph is a side remark, but we do want to avoid confusion with puff splits as well as with splits of Couette stripes (the latter actually follow yet another mechanism not only distinct from channel stripes but also distinct from puffs splits). Therefore we have now changed terminology and we refer to the splitting/breaking of localized channel stripes as 'stripe fracture'.

The question we address in Fig. 4a is if stripe fracturing is memoryless or not and we hope that the referee agrees that Fig. 4a makes it clear that it is not.

Now if we interpret the referee's remark correctly, the referee thinks this is obvious and we should see the exact same for puffs if we consider all splitting attempts of puffs (not only successful splits): This is also the case for pipe flow ($Re \sim 2200$). A puff grows in length and splits by chipping, but the downstream one does not necessarily survive as a new puff. Actually nobody has investigated this quantity for puffs, so we conducted extensive simulations for $Re=2200$ in pipe flow. The corresponding distribution of

splitting attempts of puffs, is shown on the right. The distribution is evidently exponential (Weibull type 2) and in contrast to channel stripes it is hence memoryless.

We can hence conclude that, puff splitting attempts are Weibull type 2, i.e. memoryless, stripe splitting(/fracturing) attempts are Weibull type1, i.e. aging.

We will now get to the main point the referee raised regarding splittings:

5): Regarding the referee's hypothesis that: "The success or failure of a split is memoryless."

Note that nobody has conducted any measurements or computations that either supports or disputes this statement. For our present data at $Re=710$ there are not sufficient numbers of successful reproductions to obtain resolved statistics and we have therefore carried out experiments at a larger Reynolds number of 825. This time we only consider fracturing events where the daughter stripe survives and persistently grows. We will refer to this as the "probability of stripe replication", and according to the referee this quantity should be memoryless. The result is shown in the figure on the right (the new Fig.4d in the manuscript) and as the black circles(+red curve) show this probability is evidently not memoryless. The splitting probability of puffs is shown for comparison in blue (Avila et al. 2011, their figure 3). The fit with equation 1 (red curve) identifies stripe replication as an aging process ($k=4$, see table 2). Puff splits of course exhibit a type 2, i.e. memoryless distribution as pointed out by Avila et al.

It is well understood that a single band repeats stretching and shrinking in time,

We are glad to read that the referee now agrees that channel stripes follow a cyclic process, with the change in terminology 'growth-fracturing' cycles. We can only repeat this again, puffs do not do this. Puffs as the referee in the previous report correctly stated fluctuate around a well-defined mean. We show both the fluctuation of puffs and the cyclic behavior of stripes and the corresponding distributions in the SI. There is nothing to argue here, this difference is quantified and entirely evident from SI Fig. S6. This difference between cyclic versus fluctuating is precisely the reason why puff splitting attempts are memoryless whereas the equivalent quantity for channel stripes is of type 1, aging.

but in order to examine whether decay or splitting is memoryless or not, it is necessary to observe for much longer time \$O(10^4-10^5)\$...

We also answered this point. Such long observation times have not been realized in any pipe experiment. Lifetime experiments use censored data and much shorter observation times. If channel stripes were memoryless as the referee appears to believe, then we should see decays for $Re>640$ but we don't. Again the theory and categories of lifetime distributions (equ.1) explains this. Equ. 1 actually predicts that for an aging process for $Re>640$ we should see zero decays, whereas for a memoryless process a notable fraction of stripes should decay.

...for \$Re=660\sim 700\$.

Nothing happens at $Re \sim 660 \sim 700$, there is not a single decaying case in the thousands of measurements we conducted in this regime.

I am sorry to disagree with you again and again, but I cannot accept the authors' claim, memory, based on the current experimental results.

We trust that this concern regarding the terms memory and aging is fully addressed. The theoretical analysis, is very clear on this. All our measured lifetime distributions are aging according to the definition of lifetime categories. The current experimental results could not be any clearer.

Allow us to also repeat that these are the only evidently aging (k as large as 8) lifetime distributions reported for any transitional shear flow. Much shorter-lived puffs and periodic stripes remain close to memoryless throughout (k values may slightly exceed 1 at the lowest Re but not by much).

Since this was a rather long extensive response, allow us to briefly summarize the key factual points:

(a) Based on the standard definition and categories of survival curves the lifetime distributions of localized stripes (Fig. 2b) correspond to a type1 aging process. Consequently our usage of the term 'aging' is factual and to the point.

(b) The lifetime of localized stripes can be cleanly separated from their eventual viscous decay. While the viscous decay (like for puffs) is inevitable, the lifetime is a stochastic process and follows a Weibull type 1 distribution.

(c) The characteristic time scales (τ) that we resolve for localized channel stripes are factually much longer than those of the shown, and evidently memoryless, puffs and periodic stripes. In other words, puffs however short lived remain close to memoryless, channel stripes an order of magnitude longer lived, are aging.

(d) Observation times of 10^4 - 10^5 are not needed for any other shear flow. In memoryless cases censored data provides all the information required to determine even excessive characteristic lifetimes. For aging processes data censoring is not a feasible option.

(e) The reproduction of stripes (successful 'splits', new Fig. 4d) is, per the standard definition of survival processes, a type 1 aging process.

Reviewer #4 (Remarks to the Author):

Review of Memory of transitional turbulent structures

This reviewer sees two major issues:

1. The first is language. The second reviewer and the authors' disagree primarily because language that defines and describes physical phenomena is not precise. It is used in a qualitative way. It is suggested that this can be overcome through modification of the paper. For instance, for flow physics describing aging, could be defined in a quantitative and not qualitative way near the beginning of the paper. Suggest equation(s) or mathematical properties that can be defined and applied to both the DNS and experimental measurements. For example, memorylessness should have a quantitative definition in the paper (and not referenced away) so that readers will have a firm understanding of what it is, how it is quantified, and in this case why the authors' think it is not present for the particular flow-field. This way, the ambiguity of the definition and a lack of quantification of these terms can be overcome.

Response to issue 1.:

Thank you, this is an excellent suggestion. We have meanwhile more closely looked into the theory of survival functions and it turns out that there is a single framework that defines both memoryless and aging processes. The standard way to distinguish these cases is by fitting the Weibull distribution $S(t) = e^{-(t/\tau)^k}$ to the survival data (included as equation 1 in the paper). Depending on the value of k three general types of survival curves are defined:

Type 1: $k > 0$ aging (or positive aging), the hazard rate (/probability of an event to occur) increases in time (e.g. for humans, large mammals, here the probability to die increases with age).

Type 2: $k = 0$ non-aging memoryless, the hazard rate is constant (e.g. decay of radioactive elements)

Type 3: $k < 0$ negative aging, the hazard rate decreases in time (e.g. fish, turtles etc., young have a high death rate but the longer they survive the smaller the probability they get killed).

The three types are illustrated in the left panel below.

Equation 1 allows us now to quantify if a distribution is memoryless or aging. For example, see the curves (red and black) in right panel above (new panel b of Fig 2 in the manuscript). The terminology

used is hence objectively defined by this framework (not by us) and if a process is memoryless or aging entirely follows from the fit of the Weibull distribution (equation 1). A k -value close to one (red curves in above example) defines a memoryless, non-aging process, while a value considerably larger than one (black curves) signifies an aging process.

Note that to make this distinction even clearer and to also show the trend with Re for both aging and non-aging cases, we added the additional data sets for puffs (red data in above figure, from Avila et al. 2010) and one example for non-localized, periodic stripes (Fig. 2c, taken from Gomé et al. 2020). All the data sets shown are fitted with the Weibull distribution (equ.1) and the fit parameters are given in table 1 so that the differences between the types of curves are quantified.

Hence, as requested, we remove any ambiguity and all the terms used are based on the standard definition. Moreover the fits to our data, see right panel above, match the data near perfectly.

Finally the distinction between these aging processes and the memoryless cases is quantified in the newly added tables 1 and 2. These tables provide all the fit parameters and clarify that:

- a) there is a clear separation between aging and non-aging processes (see values of k , the shape factor), which makes this distinction unambiguous;
- b) the characteristic lifetimes we resolve for localized channel stripes are long. Sometimes even by an order of magnitude longer than the memoryless puff lifetimes we compare to (see values of τ).

Two additional remarks:

- Regarding, 'why the authors' think [memorylessness] is not present for the particular flow-field': First of all we hope the analysis clarifies that memorylessness is indeed absent. Regarding the 'why', as we point out in the manuscript, aging is associated with the large scale flow around the stripe (SI Fig. S5) and with the tip instability (Fig.3), that this large scale flow causes. It is noteworthy that this instability is not present in other flows and specific to fully localized channel stripes. The slow-down of the large scale flow is shown in SI Fig. 5 and the weakening of the tip activity is demonstrated in Fig. 3b (decrease of the tip velocity).

The important point this decomposition makes is that, in contrast to the aging large scales, the small scales remain intact. We are now showing this result in the main part of the paper, in Fig. 2d (energy density curves in colour in lower panel). These small scales correspond to the turbulence in the stripe's interior, essentially streaks and vortices. Based on this result the stripe's lifetime, corresponding to the plateau, can be cleanly separated from the stripe's eventual demise, the latter being the eventual viscous decay. This decomposition had been requested by an earlier referee. Conversely referee 2 appears to not have realized this distinction between the lifetime and the eventual stripe decay. Hence we explain this in more detail in the new version of the manuscript.

Another important insight from this decomposition is that after the departure from the energy plateau, the viscous decay process takes about 200 advective time units. This viscous decay time happens to be the same for the aging localized stripes (in colour) and the periodic memoryless

stripes in small domains (in grey). All of this means that a clear distinction can be made between the life and the eventual death of the stripe and hence that the lifetimes are well-defined, just as they are for the memoryless cases.

- There has been one additional issue with terminology in the exchanges with referee 2, which is with respect to the term splitting. Our definition of a split is the literal one, i.e. that the stripe splits into two pieces and our emphasis in Fig. 4 a-c is on the cyclic dynamics of the parent stripe. Referee 2 apparently misunderstood this and expected splittings (like they are defined for puffs) to only count if the daughter stripe survives.

To address this and to avoid confusion with puffs splits, we have now changed the terminology. We call the event of a stripe splitting into two pieces “stripe fracturing” whereas we refer to the proliferation event, i.e. those incidences where following a stripe fracture, the newly created daughter stripe grows and survives, as “stripe reproduction”. For the former process we have added the corresponding Weibull fit to Fig. 4a.

Regarding the latter stripe reproduction, and this was another main critique by referee 2 we added new experiments, **new Fig. 4d**. While referee 2 claimed that this production of new stripes would be a memoryless process, the data is very clear on this. Again we fitted equation 1 and the terminology follows from the value of k . According to the lifetime framework this is evidently not memoryless but an aging process. Again also from Fig. 4d the fundamental difference between the reproduction of localized stripes and the memoryless case of puff reproductions hopefully now is clear and quantitatively confirmed (also see table 2).

2. Second, comes from my own review of the paper. Multiple points regarding what are new are buried within large sections of text (without paragraph breaks). It is suggested that in the introduction of the paper that the authors' are very a) clear, b) up front, c) explicit, and d) quantitative about the claims they are making in the paper. These statements are present, but the reviewer had to work hard to find them.

Response to issue 2.:

We also agree on this second point. The text has become rather cluttered after three rounds of review. We have made an effort to remedy this and rewrote the introduction while also including the referee's suggestions a) to d). Specifically we also changed the last paragraph of the introduction. At the same time we implemented the 'specific comments', see below. In addition we added subheadings to the manuscript which hopefully provide a much clearer structure.

Otherwise, the points back and forth between reviewer 2 and the authors' are reasonable, and the reviewer believes comes from the fact that reviewer 2 and the authors' essentially have different viewpoints or definitions of qualitative wording to describe a quantitative subject. In science and math, it is advisable to be as quantitative and explicit with definitions as possible. This would avoid this entire situation. Then the paper can be assessed with the same basis and understanding.

By making these two changes, the reviewer believes that the disagreements between authors and previous reviewers can be resolved.

Specific comments on Writing in Revised Article

For a manuscript of this length, there is an overly lengthy introduction through pages 1 through half of 4. The reviewer has no idea why all this introduction material is important until the author states what they are doing midway through page 4. It would be better to inform the reader what the intention of the paper is in the beginning. Also see general comment 2. Major implications of the paper are spread too far apart and are buried in particular paragraphs.

The introduction indeed contained various points that in hindsight are not as essential anymore. We have removed or shortened these parts and when necessary moved discussion points to the later parts of the paper, when they become relevant. Overall this shortened the introduction by around one page. As mentioned above, we have also revised the last paragraph of the introduction which now more precisely states what will be shown.

Page 5 – Awkward to have a single sentence as a paragraph.

Corrected

Page 5 – Spacing between turbulence , - e.g. no space.

Ok

Page 5- spacing between (and stripe

Ok

Page 5 – Should have paragraph break near end before We propose... Suggest moving this core idea earlier as it is buried within paper and an important point.

Ok

Page 7 – Buried comment regarding research performed regarding DNS that was not mentioned explicitly prior.

We now created a separate subsection for the DNS.

Page 8 to 10 – Please use paragraph breaks in your writing to separate core ideas. This is fundamental in technical writing. It makes it difficult to understand core ideas in your writing.

Ok we have tried to follow this recommendation.

Noting early on that some spacing is not made between references and words.

Ok.

SI Page 6 last line – undefined reference with ? .

Note: It would be appreciated to have line numbers on manuscripts to review for easier reference. Have been added.

Overall we would like to thank the referee again for these very helpful and constructive comments and we think that the data, now being embedded in the correct theoretical framework, speaks for itself. In particular the Weibull fits in Figures 2 b,c and 4 d make the distinction between aging and non-aging process unambiguous and quantitative.

Response to referees:

Referee 5:

We thank the referee for carefully reviewing our manuscript and we appreciate the constructive suggestions:

1. Stripe length is a key parameter for this study, but is not defined in both the manuscript and the supplementary file, and should be added in the new version. Is it the length in the oblique direction? the streamwise direction? or the spanwise direction? How is it determined in the image processing and in the numerical simulations?

This information has indeed been missing. We have now added a section in the supplementary material, where we describe how the stripe length is determined in the image processing. As explained there, the stripe length is the actual length, which also takes into account any slight curvature that the stripe may have. It is hence neither a projection onto the streamwise, spanwise nor on the oblique direction. Note that we do not calculate or use the stripe length in the simulations, instead we are using a cutoff on the kinetic energy of the stripes to determine the decay time.

2. It has been shown that the DP analogy is applicable for the stripes in channel flows as Re is larger than 1000 (e.g. Phys. Rev. Fluids, 113902, 2019), so the statement in the abstract “the one to one mapping between turbulent stripes and active DP-sites is not fulfilled” seems arbitrary. Different from most individual stripes as $Re > 1000$, the stripes at low Re are isolated turbulent ones: owing a clear downstream head and an upstream tail, and this is the reason why the flow state at the low Reynolds number range is referred to as the sparse turbulence, where the DP analogy is not applicable (Phys. Rev. Fluids, 011902, 2018). Therefore, it is suggested to emphasize that the analyses and statements are made for the isolated turbulent stripes at low Reynolds numbers or at the sparse turbulent state.

Thanks for this comment, and in accordance with the referee’s suggestion we are now emphasizing that the regime we are dealing with is indeed the isolated stripe regime. Naturally there is only one critical point, at which the turbulent fraction goes to zero, and for channel flow in this limit indeed individual stripes are isolated. In accordance with the referee’s remark we have changed the abstract as follows: ‘As demonstrated by a standard analysis of the respective survival curves, **isolated** channel stripes, **in the immediate vicinity of the critical point**, age.’ In the following sentence we added ‘**in this low Reynolds number regime**’.

3. The initial conditions, e.g. the initial kinetic energy in DNS or the initial length of the stripe, have substantial effect on the lifetime and the growth rate of stripes: longer initial stripes may have larger growth rate in a longer period before their final decay (see Phys. Fluids, 041702, 2015). Consequently, some turbulent stripes may leave the outlet of the channel in experiments before starting their decay process and are identified as growth cases, leading to an underestimation of the critical Reynolds number. We understand that the experimental channel cannot be infinitely long, but it is suggested to add figures, showing the length (or kinetic energy) distribution of the initial stripes, and the corresponding discussions.

Indeed as pointed out by the referee, Xiong and co-workers (Phys. Fluids, 041702, 2015) reported a dependence of the growth rate on the initial stripe energy. This applies to short (i.e. low kinetic energy) stripes, whereas above a critical kinetic energy (ℓ /length) stripes persistently grow. The answer to the referee's question is very straightforward, the stripes initiated in our experiment are all above this critical energy level. Hence in full agreement with Tao et al. we do not see any dependence on the initial stripe length.

To provide some more details:

Xiong et al. give this energy level in their Figure 4c (grey bar), corresponding to 0.001 in dimensionless units. While they don't give the exact corresponding length, it can be estimated from the length of the stripe shown in Figure 3c combined with the corresponding energy level in Figure 3a. According to this, 0.001 approximately corresponds to a stripe length of 110h.

Our initial stripe lengths distributions have now been added, as also requested by the referee, as Fig.S3 in the supplementary materials. As this figure shows, with 180h the mean initial stripe length in our experiment is well above the threshold identified by Xiong and co-workers.

In accordance with this, please also note that our Fig. 4c shows that the growth rate of our stripes is constant. Independent of their initial lengths, all stripes follow the same diagonal. Given that in our measurement lengths are greater than 150h, this is perfectly in line with Xiong et al's observations.

4. The second key point in the abstract is about the chaotic saddle concept, but this concept is not mentioned at all in the manuscript and the supplementary file except the introduction part, and hence it is suggested to add some words explaining the relation between the memoryless decay and the chaotic saddle.

Thanks for pointing this out! We have now added a clarifying statement on lines 21-22: 'a chaotic saddle in phase space, characterized by an exponential lifetime distribution[1], i.e. a constant escape rate[2], independent of age. This correspondence to a chaotic saddle is probably the best established property of transitional turbulence in subcritical shear flows,...'

We here refer to a paper by Faisst and Eckhardt[2], where they list an exponential lifetime distribution as a characteristic signatures of chaotic saddles.

Referee 6:

We thank the referee and we are particularly grateful to the referee for taking the time to look at the exchange with the previous referees 2 and 4 and to provide a neutral scientific assessment. We fully agree with the entire assessment.

With respect to the referee's additional remarks:

In the abstract "The discrepancy between channel flow and the established transition model... "

It would be more precise and clearer to say more explicitly, "... between channel flow and the transition models established for Couette flow, pipe flow..."

We have changed this sentence as suggested.

Line 177, please clarify, including fully localised stripes in Couette flow?

Indeed fluctuation of energy around a well-defined mean is expected for all memoryless cases including

fully localized Couette stripes. We have reworded the sentence (now on line 181) to clarify this point: 'This is qualitatively different from memoryless cases, such as puffs or localized stripes in Couette flow[42], where the energy is expected to fluctuate around a constant mean...'